# Turn-key mapping of cell receptor force orientation and magnitude using a commercial structured illumination microscope

Aaron Blanchard[1], J. Dale Combs[2], Joshua M. Brockman[1], Anna V. Kellner [1], Roxanne Glazier[1], Hanquan Su[2], Rachel L. Bender [2], Alisina S. Bazrafshan[2], Wenchun Chen[3,4], M. Edward Quach [3,4], Renhao Li [3,4], Alexa L. Mattheyses [5] & Khalid Salaita [1,2✉]

Many cellular processes, including cell division, development, and cell migration require spatially and temporally coordinated forces transduced by cell-surface receptors. Nucleic acid-based molecular tension probes allow one to visualize the piconewton (pN) forces applied by these receptors. Building on this technology, we recently developed molecular force microscopy (MFM) which uses fluorescence polarization to map receptor force orientation with diffraction-limited resolution (~250 nm). Here, we show that structured illumination microscopy (SIM), a super-resolution technique, can be used to perform super-resolution MFM. Using SIM-MFM, we generate the highest resolution maps of both the magnitude and orientation of the pN traction forces applied by cells. We apply SIM-MFM to map platelet and fibroblast integrin forces, as well as T cell receptor forces. Using SIM-MFM, we show that platelet traction force alignment occurs on a longer timescale than adhesion. Importantly, SIM-MFM can be implemented on any standard SIM microscope without hardware modifications.

[1] Wallace H. Coulter Department of Biomedical Engineering, Georgia Institute of Technology and Emory University, Atlanta, GA, USA. [2] Department of Chemistry, Emory University, Atlanta, GA, USA. [3] Aflac Cancer and Blood Disorders Center, Children's Healthcare of Atlanta, Atlanta, GA, USA. [4] Department of Pediatrics, Emory University School of Medicine, Atlanta, GA, USA. [5] Department of Cell, Developmental, and Integrative Biology, University of Alabama at Birmingham, Birmingham, AL, USA. ✉email: k.salaita@emory.edu

Cellular forces are generated by the cytoskeleton and then transmitted through membrane receptors to other cells and the extracellular matrix. These pN forces are critical for maintaining a wide variety of essential processes in mammalian cells, including development, migration, immune recognition, and coagulation[1–7]. Over nearly a decade, our group has developed an array of molecular tension fluorescence microscopy (MTFM) probes for quantifying and visualizing the pN-scale molecular forces transmitted by cell-surface receptors[4,8–19]. These probes consist of an extensible "molecular spring" flanked by a fluorophore–quencher pair, and are generally conjugated to a substrate such as a glass coverslip[8,10,11,14,16,20,21] or lipid membrane[22,23]. The MTFM probes present peptide or protein ligands that are recognized by cell receptors (e.g., integrins). When the cellular receptor applies sufficient force, the MTFM probe will unfold and extend, resulting in dequenching of the fluorophore and a marked increase in fluorescence. This mechanical signal can thus be imaged in space and time using conventional fluorescence microscopy (Fig. 1a).

Compared to traction force microscopy (TFM), the gold standard technique for measuring cellular tractions, MTFM offers a significant improvement in spatial resolution (~0.25 μm versus ~μm). MTFM signal also provides information about the characteristics of pN forces transmitted through individual receptors, rather than the collective nN forces generated by the cell[8]. Hence, MTFM is attracting the interest of the cell biology community[24–26]. Nonetheless, an important distinction between MTFM and TFM is that, while TFM is used to measure both force

magnitude and force orientation, MTFM only reports on force magnitude. This is indeed a limitation, as recent work has increasingly shown that force orientation is transduced into biochemical signals. For example, the T-cell receptor (TCR), integrins, and vinculin have been shown to be orientation-dependent mechanosensors[3,4,27]. In addition to orientation, recent super-resolution work has shown the importance of nanoscale organization of force-bearing structures[18,28]. Many physiologically important force-bearing structures—such as focal adhesions[8], lamellipodial protrusions[4], T-cell receptor clusters[17], and podosomes[23,29]—are hundreds or tens of nanometers (nm) in their smallest dimension and also generate highly directional forces. Thus, tools for super-resolution MTFM that also report on force orientation are highly desirable.

We recently took advantage of fluorescence polarization imaging to report on the force orientation of molecular probes[16]. This approach, dubbed molecular force microscopy (MFM), is enabled by the combination of DNA mechanotechnology[30] tension probes – which include DNA hairpins as their "molecular springs"—with excitation-resolved fluorescence polarization microscopy[31–41]. DNA hairpins are well-suited for molecular force measurement because they offer a nearly-"digital" force response; a DNA hairpin can exist in one of two states—folded or unfolded—and the fraction of time spent in the folded state decreases from ~95% to ~5% over a narrow force range of 2–3 piconewtons (pN) centered around the probe's $F_{1/2}$. The $F_{1/2}$ value, defined as the force at which the probe spends equal time in each of the two states, can be tuned from ~2 to ~19 pN by adjusting the length and GC content of the hairpin[42]. Due to highly efficient quenching, the fluorescence intensity of the opened state is ~20–100× brighter than that of the closed state[8].

The fluorophore most commonly used in DNA hairpin probes, cyanine 3B (Cy3B), has two properties that make MFM possible: First, Cy3B, like most fluorophores, has a transition dipole moment that can be approximated as a 3-dimensional (3D) vector, $\mu$ (Fig. 1b)[16,43]. (The transition dipole moments for excitation and emission are highly similar for Cy3B, and can thus be approximated as a single vector[16]). The intensity ($I$) of a fluorophore excited by plane-polarized excitation light (which has a fixed polarization vector denoted by $E$) can be described by the relationship:

$$I \propto \langle \cos^2(\psi) \rangle \tag{1}$$

where $\psi$ is the angle between $\mu$ and $E$, and $\langle \rangle$ brackets denote ensemble averaging across the probability distribution of orientations that $\mu$ and $E$ adopt over time. When $E$ is rotated (e.g., by placing a rotatable half wave plate in the path of the excitation beam) and multiple images of a sample are taken with different fixed $E$ orientations, the set of $I$ values recorded for each pixel can be used to calculate the average fluorophore orientation in the physical region corresponding to that pixel.

Second, when Cy3B is covalently coupled to the terminus of a DNA duplex, it inherently stacks perpendicular to the duplex's long axis[44]. This property, and the nature of our tension probe design, cause Cy3B to align itself such that $\mu$ is perpendicular to the long axis of the opened probe (Fig. 1b). Because probe orientation is controlled by receptor-ligand force orientation, $\mu$ is therefore perpendicular to the force vector $F$. We assume that the probe can freely rotate around its long axis thanks to the incorporation of flexible linkers flanking the probe. Accordingly, $\mu$ can take an ensemble "disk" of orientations perpendicular to $F$ (Fig. 1b). In this scenario, fluorescence intensity can be described by the equation:

$$I = A \sin^2(\alpha - \phi) + c \tag{2}$$

where $\phi$ and $\alpha$ are the in-plane angles of the $F$ and $E$, respectively

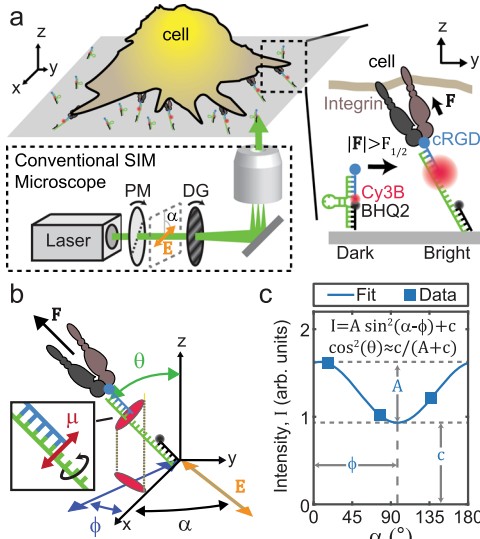

**Fig. 1 SIM-MFM concept. a** Diagram showing microscope setup and DNA hairpin-based tension probe function. We show the basic components of a SIM microscope excitation line: a linearly polarized laser passes through a polarization modulator (PM) and a diffraction grating (DG). These two components rotate together such that α (the angle of the laser beam's electric field vector, **E**) is the same as the orientation of the striping pattern created by the DG. Inset shows a DNA hairpin-based tension probe (blue, green, and black DNA strands) transitioning from a closed state to an opened state upon application of tension (**F**) with a magnitude exceeding the probe's $F_{1/2}$ value. The cell receptor (brown and gray) is an integrin and the ligand (blue circle) is cRGD. **b** Illustration of **F** (black arrow) and **E** (orange dipole) angles used in this work. Inset shows Cy3B transition dipole moment (**μ**) perpendicular to the long axis of the tension probe. **c** Representative data (blue squares) and curve fit (blue curve) from a pixel indicated in Fig. 2c with a green arrow. A, c, and φ are illustrated and the equations used to obtain curve fit from data are shown.

(Fig. 1b) and $A$ and $c$ are the sinusoid's amplitude and offset and depend on factors such as the tilt angle of the force, $\theta$ (Fig. 1b), and the surface density of opened tension probes. Note that $\alpha$ is the independent, user-controlled variable while $\phi$, $c$, and $A$ are the parameters being measured. We previously derived[16] an equation to roughly calculate $\theta$:

$$\theta = \cos^{-1}\left(\sqrt{\frac{\frac{c}{A+c} - b}{1 - b}}\right) \qquad (3)$$

where $b$—which we previously estimated to be 0.069 – is used to account for the estimate that the fluorophore spends ~10% of its time unstacked from the duplex terminus (during which time the fluorophore is randomly oriented)[44]. Note that the 180° periodicity, as shown in Eq. (2), means that force orientation is degenerate (i.e., any orientation measurement is indistinguishable from the same orientation rotated 180° around the $z$ axis). This two-fold degeneracy in force orientation measurement is a fundamental limitation of MFM that could potentially be addressed in the future using inclined illumination approaches[41].

In our initial implementation of MFM[16], we utilized a rotating half wave plate to vary $\alpha$ continuously and recorded a series of 73 images over the course of ~3.2 s. We then used least-squares residuals fitting to fit the intensity values for each pixel to Eq. (2). Currently, MFM requires a microscope setup where $\alpha$ can be modulated freely. However, this is not a standard feature in the vast majority of commercially available fluorescence microscopes. As a result, MFM is technologically inaccessible to a large proportion of the cell biology community. However, because Eq. (2) has three unknown fit parameters ($\phi$, $\theta$, and $A$), three unique $\alpha$ values (e.g., $\alpha = 0°$, $60°$, and $120°$) would technically be sufficient to obtain a unique best-fit curve analytically (Fig. 1c).

As stated above, enhancing the spatial resolution of force mapping is desirable due to the nanoscale size of many mechanically active structures of interest. Currently, the spatial resolution of MTFM-based techniques is limited by the diffraction limit of light, which spreads the detected fluorescent signal from a single fluorophore out over a spot ~250 nm in diameter ($\lambda/2NA$). While we recently developed a method for localization of integrin forces with a spatial resolution of ~20 nm, this method does not capture force orientation information[18]. Similarly, while protein-based force probes have recently been used for super-resolution force mapping, such probes have not yet been shown to be capable of reporting force orientation information[45–49]. Super-resolution imaging has been employed to improve the spatial resolution of TFM[50–52], but even under ideal theoretical circumstances, the spatial resolution is not expected to exceed ~500 nm[51]. This constraint arises because of TFM's inherent need for substrate deformation, as well as the optical diffraction limit (see Supplementary Table 2 for a comparison of high-resolution TFM and MTFM techniques). Accordingly, improving the spatial resolution of MFM would enable measurements of force orientation with unprecedented resolution.

Recently, Zhanghao et al. reported a technique called polarized structured illumination microscopy (pSIM), which can be used to map fluorophore orientation with a high spatial resolution (~100 nm) and fast temporal resolution (<1 s)[53]. Inspired by this recent development, here we report the successful adaptation of pSIM to MFM[16] to generate ~110 nm resolution maps of pN-scale forces generated by living cells, including force orientation information. We call this technique SIM-MFM. SIM is an increasingly utilized super-resolution fluorescence imaging modality that achieves ~100 nm spatial resolution by exciting fluorescent samples with a structured pattern of excitation light. Many research institutions maintain SIM microscopes within core facilities or within a small number of research labs. Importantly, SIM-MFM can be implemented in a turn-key fashion using commercially available structured illumination microscopes without any hardware or software modifications. With this advance, we simultaneously improve the spatial resolution and the accessibility of MFM with a commercially available microscope.

## Results

**Proof-of-concept demonstration of SIM-MFM with human platelets.** The SIM technique fundamentally depends on polarization modulation, which conveniently coincides with the optical requirements for performing MFM (Fig. 1a). Specifically, SIM[54–57] works by using a diffraction grating to create an excitation beam illumination pattern that varies sinusoidally in one direction. The sinusoidal interference pattern is then phase-shifted back and forth and images are acquired at multiple (generally a total of three or five) distinct phase shifts. The orientation of the illumination pattern is then rotated and this process is repeated at two or more distinct illumination pattern angles, resulting in a minimum of nine images (if three-phase shifts per stripe orientation are used) or fifteen images (if five phase shifts per stripe orientation are used). These images are then processed using a reconstruction algorithm that produces a single image with a spatial resolution that is (at best) double what could be achieved using widefield epifluorescence. Importantly, the sinusoidal diffraction pattern can only form properly when the excitation light is linearly polarized, with $E$ parallel to the striping orientation. Therefore, when the striping orientation is modulated, so is $\alpha$ (Fig. 1a). Accordingly, polarization modulation is a fundamental and enabling characteristic of SIM.

To test whether SIM could be used to implement MFM, we first performed experiments to validate our commercially available Nikon Eclipse Ti-based SIM microscope (N-SIM). First, we used a photometer and a linear polarizer to verify that the microscope's excitation laser was linearly polarized (Supplementary Fig. 1). Second, we imaged microspheres coated with tetramethylindocarbocyanine (DiI)-doped supported lipid bilayers. DiI is a cyanine dye (like Cy3B) linked to two lipid tails that spontaneously insert into supported lipid bilayers such that $\mu$ is roughly parallel to the microsphere surface. As expected, we observed a sinusoidal variation in $I$ around the perimeter of the microsphere, with maximal intensity corresponding to the spot where $\mu$ was aligned with $E$ (Supplementary Fig. 2). As $\alpha$ varied during each SIM acquisition, the sinusoid shifted by a commensurate amount, thus confirming that the polarization modulates with the direction of the stripe pattern for our commercial SIM microscope at an interval of ~60°. Each full set of images was collected in ~1 s—notably more rapid than the previous implementation of MFM[16] which required ~3.6 s per acquisition.

We next tested whether SIM could be used to implement MFM. To do this, we prepared a typical DNA hairpin-based MTFM experiment by conjugating DNA tension probes ($F_{1/2} = 4.7$ pN) presenting cyclic arginine-glycine-aspartic acid-D-phenylalanine-lysine (cRGDfK, a peptide that binds $\alpha_v\beta_3$ and $\alpha_5\beta_1$ integrin receptors with high affinity[58]) to a glass coverslip through biotin-streptavidin binding. Previous surface density calibration of surfaces prepared in this manner[16] revealed a density of ~700 tension probes per $\mu m^2$. We next deposited quiescent human platelets on these surfaces. We used platelets as a model because they are small and highly mechanically active, and because integrin mechanics are important in platelet physiology[59]. Indeed, resolving the spatial distribution of mechanical platelet traction forces is very challenging for state-of-the-art traction force microscopy due to platelets' small size (~2–5 μm spread diameter)[4,16,60–62].

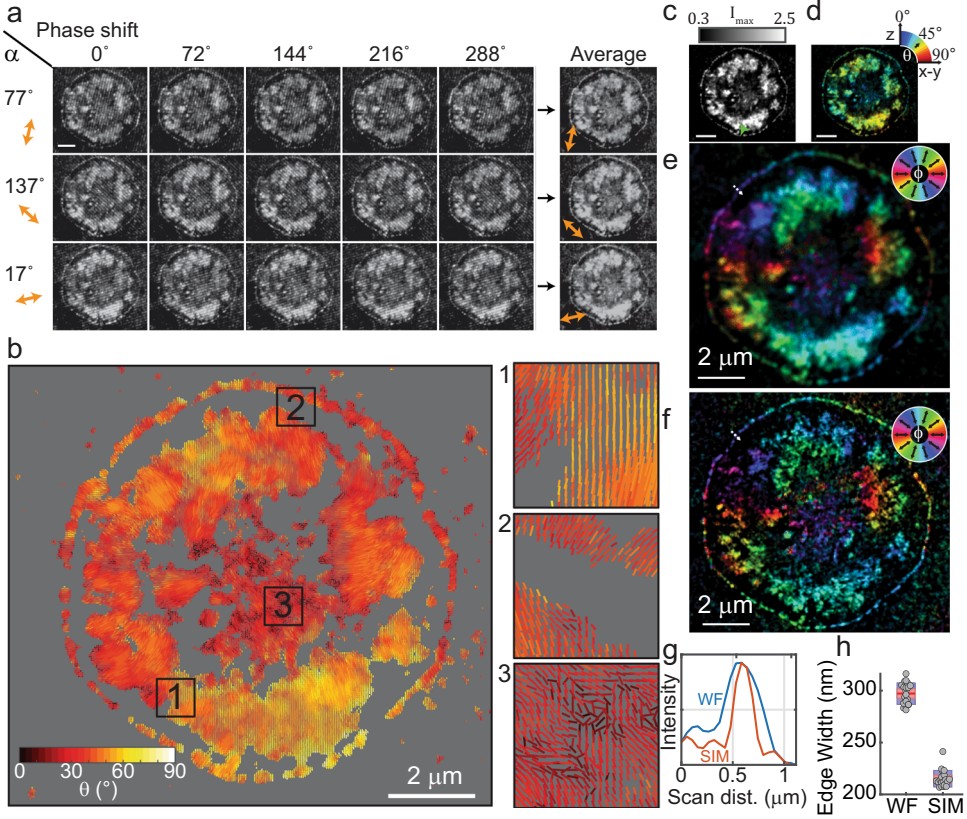

**Fig. 2 MFM can be performed on an unmodified SIM microscope. a** Representative raw images of a platelet imaged using SIM-MFM. Fifteen images—one at each of 5 phase shifts and 3 illumination pattern orientations – are shown. The sinusoidal diffraction pattern can be seen upon careful inspection. Scale bar is 2 µm in all images throughout this figure. Images were contrast-enhanced to make the striping pattern more apparent. Orange dipoles denote the approximate orientation of the excitation polarization for images in each row. **b** Dipole orientation map of the platelet, wherein each pixel with $I_{max}$ greater than a threshold (in this case, the 60th quantile of the image) has a dipole associated with it. For each dipole, $\phi$ is encoded in the dipole's orientation, $\theta$ is encoded in the dipole's color (see colormap in the bottom left of image), and $I_{max}$ is encoded in the dipole's length. Insets 1–3 show zoom-ins on specific regions of the dipole map, which are denoted with black boxes and corresponding black number labels. This display method was developed in our previous work[16]. **c–e** Alternative colormap-based display style used in this work showing **c** $I_{max}$ (normalized to the background, as depicted using colorbar above the image), **d** $\theta$, and **e** $\phi$ for each pixel. For **d**, **e**, the color reflects the measured force orientation, as depicted using color wheels on the top-right of each image. For all three colormaps, the brightness of each pixel is scaled to the $I_{max}$ value shown in **c**. The images were also contrast adjusted such that the brightness is bounded to the 35th and 99th percentiles of the pixels' $I_{max}$ values. **f** Super-resolution reconstruction of the platelet. **g** A linescan (dotted white arrows in **e** and **f**) of the background-subtracted and normalized intensity of a platelet's lamellipodial edge for both widefield (WF) and super-resolution (SR), showing an enhancement in spatial resolution when using the SIM reconstruction. **h** Boxplots showing the median width of 17 platelets' lamellipodial edges, as measured in WF and SR (Supplementary Fig. 6). Solid red line indicates mean, dashed red line indicates median, red boxes show the standard error of the mean, and blue boxes show 95% confidence interval of the mean. Gray circles indicate individual data points. Independent experiments with human platelets were repeated five times with similar results.

Within ~10–20 min of seeding, platelets engaged and spread on the surface. Association between platelet integrin receptors and cRGDfK ligands promoted outside-in integrin activation and initiated platelet signaling, spreading, and cytoskeletal remodeling. These processes then led to the transmission of forces through the integrin-ligand bonds and opening of the tension probes[4]. We collected SIM images of well-spread platelets within ~45 min. Typically, platelets spread out over an area ~2–5 µm in width and their MTFM maps generally exhibited two distinct features: an outer ring of tension, which coincides with actin polymerization at the lamellipodial edge of the cell, and a large inner region with lobes of tension, which arises from actomyosin activity in the body of the platelet. The tension patterns observed here were consistent with those observed in our prior work[4,16,18] validating the rigor and reproducibility of the probe and the biological model (Figs. 2, 4 and Supplementary Fig. 3).

We next tested whether the polarization modulation of the SIM microscope could be used to measure force orientation. For

our initial assessment, we used the built-in "3D-SIM" function of our N-SIM microscope to acquire three sets of five images (Fig. 2a—five phase shifts at each of three striping orientations). We performed illumination correction using images of either a chroma slide or a surface with unquenched tension probes taken on the same day. We found that both methods performed similarly, but the chroma slide method was much less labor-intensive (Supplementary Fig. 5). We then simply took the average of the images acquired at each striping orientation, which is mathematically valid due to the specific phase shifts used in SIM. This process results in a total of three images, each corresponding to a distinct $\alpha$ value (Fig. 1c). We then loaded the image into fairSIM[63] (an open-source ImageJ plugin that enables facile analysis of SIM data), which revealed striping orientations (and therefore $\alpha$ values) of 77°, 137°, and 17°. Accordingly, for a given pixel, we designate the three respective intensity averages as $I_{17°}$, $I_{77°}$, and $I_{137°}$. Rather than applying least-squares residuals fitting to the $I$ vs. $\alpha$ curve as performed previously[16],

we derived analytical equations (Supplementary Note 1) to rapidly obtain a unique fit to Eq. (2) for each pixel:

$$\phi = \frac{1}{2}\tan^{-1}\left(\frac{I_{avg} - I_{137°} - \left(I_{avg} - I_{77°}\right)\cos(120°)}{\left(I_{avg} - I_{137°}\right)\sin(120°)}\right) + 77° \quad (4)$$

$$A = \frac{2\left(I_{avg} - I_{137°}\right)}{\cos\left(2\left(77° - \phi\right)\right)} \quad (5)$$

$$c = I_{77°} - A\sin^2\left(77° - \phi\right) \quad (6)$$

$$\theta = \cos^{-1}\left(\sqrt{\frac{\frac{I_{max}}{c} - 0.069}{1 - 0.069}}\right) \quad (7)$$

Where $I_{avg}$ is the average of $I_{17°}$, $I_{77°}$, and $I_{137°}$. Finally, $I_{max}$ is the maximum intensity of the fit sinusoid:

$$I_{max} = A + c \quad (8)$$

Note that $I_{max}$ is a better metric for brightness than $I_{avg}$ because $I_{max}$ is (very nearly) proportional to the number of opened probes regardless of force orientation. We then applied these calculations to the entire image set. Note that Supplementary Note 1 includes calculations that are generalizable to SIM systems with different sets of $\alpha$ triplets, so long as the $\alpha$ values are separated by 60°. Our initial results reproduced the force patterns that are expected for platelets, demonstrating that SIM-MFM is viable (Fig. 2b–e).

We utilized two rendering methods for displaying orientation data, both of which are shown in Fig. 2. Our first display method simultaneously shows $I_{max}$, $\phi$, and $\theta$. We mask the image to the 60th percentile intensity of the image and, for each pixel, plot a dipole (on a gray background) with orientation related to $\phi$, length proportional to $I_{max}$, and color related to $\theta$ (Fig. 2b) (a percentile in this work refers to the intensity level that is brighter than that percent of pixels in the image). Our second display method is a colormap-based approach that gains some spatial detail at the expense of only being able to encode $\theta$ or $\phi$ in a single image[53]. In this display method, orientation is linked to pixel color (using the circular HSV colormap for $\phi$ or the jet colormap for $\theta$) and pixel brightness is linked to $I_{max}$ (Fig. 2c–e and Supplementary Fig. 3). For clarity, we scaled the minimum and maximum pixel brightness to the 35th and 99th percentiles, respectively, of the entire image.

The platelets that we imaged with SIM-MFM displayed many characteristics that we observed in our original MFM study[16]. The platelets display two distinct regions of tension; at the lamellipodial edge and the inner lobe(s). As previously observed, the lamellipodial edges generally display a highly isotropic radial pattern (i.e., $\phi$ values are generally perpendicular to the platelet's edge and point towards the centroid of the platelet) while the platelets' inner regions generally display (with a few exceptions) two to four "lobes" that appear to act as independent and internally-homogenous mechanical units. For example, Supplementary Fig. 4 shows a platelet with two lobes in its inner region and a clearly visible lamellipodial edge. While the outer ring of lamellipodial tension points isotropically inward, the inner lobes display highly anisotropic contraction, with all dipoles within the lobes generally pointing to the central axis that bisects them. Several platelets' $\phi$ maps are shown in Supplementary Fig. 3. SIM-MFM also faithfully reproduces previously recorded $\theta$ values; an analysis of $\theta$ values across many platelets ($n = 37$ platelets) also revealed that the average $\theta$ value, $\langle\theta\rangle$, is $47 \pm 3°$. This

measurement is similar to the value of $39 \pm 2°$ that we measured previously[16] and the ~$50 - 60°$ measurements of the tilt angle of the adaptor protein talin[28,64].

**Super-resolution reconstruction and two-color imaging of GFP-paxillin-expressing 3T3 fibroblasts**. We next leveraged the inherent super-resolution capability of the SIM technique by processing platelet SIM-MFM acquisitions using the Nikon Elements software associated with our N-SIM microscope. Following precedent from the original demonstration of pSIM by Zhanghao et al.[53], we then used the $\phi$ map calculated from the diffraction-limited images (as described above) to color the super-resolution intensity maps (Fig. 2f). Fundamental limitations of the pSIM technique (see ref. [53] for details) prevent direct super-resolution mapping of molecular force orientation, but the pseudo-super-resolution molecular force orientation maps revealed by this process have the potential to uncover new information about the nanoscale organization of molecular forces.

To estimate the resolution enhancement offered by SIM reconstructions, we employed a linescan-based method[65] to estimate the thickness of the tension signal produced by platelets' lamellipodial edges (Fig. 2g). Our systematic analysis of 17 platelets revealed a consistent enhancement in spatial resolution of ~82 nm (Fig. 2h and Supplementary Fig. 6). This resolution enhancement is close to the theoretical limit of ~95 nm (see Supplementary Fig. 6 for more detailed description), thus validating the quality of the data and reconstruction process. As a second means of directly quantifying resolution, we used parameter-free image resolution analysis software[66]. We ran 12 pairs of SIM-MFM images (widefield and super-resolution) through this software and observed a spatial resolution of $193 \pm 8$ nm in widefield and $114 \pm 2$ nm in super-resolution (Supplementary Fig. 7).

We next used SIM-MFM to image integrin force orientation in a 3T3 mouse fibroblast cell line stably transfected with GFP-tagged paxillin (a focal adhesion protein). Integrin forces in this cell line are primarily localized to focal adhesions (FAs)[16], which are 10–100 s of nm wide and 100–1000 s nm long and transmit contractile forces from the cell to the surface[28,67,68]. Our results using the SIM system reproduced these previous findings, revealing FA force orientations that generally pointed toward the cell centroid (Fig. 3). Figure 3a (top) shows $\phi$-colored diffraction-limited and super-resolution maps, while a $\theta$-colored map (Fig. 3b) shows tilt angles close to 45°, as demonstrated previously[16]. We also performed SIM acquisitions of GFP-Paxillin as shown in Fig. 3a (bottom). Super-resolution reconstruction of the GFP-Paxillin image demonstrates an added benefit using the SIM-MFM approach: that nanoscale molecular organization and molecular force orientation and organization can be mapped in super-resolution concurrently with high temporal resolution. The super-resolution reconstructions of these images reveal a common pattern: many regions of tension that present as individual FAs using diffraction-limited MFM are present as multiple smaller structures with distinct mechanics (Fig. 3e, f). The observation of these sub-regions within FAs, which has been borne out in the previous literature[28,69,70], can also be seen in additional examples in Supplementary Fig. 8. We next evaluated our SIM acquisitions using the ImageJ plugin SIMcheck[71], which rates the quality of SIM data via objective control parameters. This analysis, shown in Supplementary Fig. 9, indicated that our acquisitions were "usable", and "low-to-moderate" quality. The limitations in image quality, as well as potential future directions, are addressed in greater detail in the discussion.

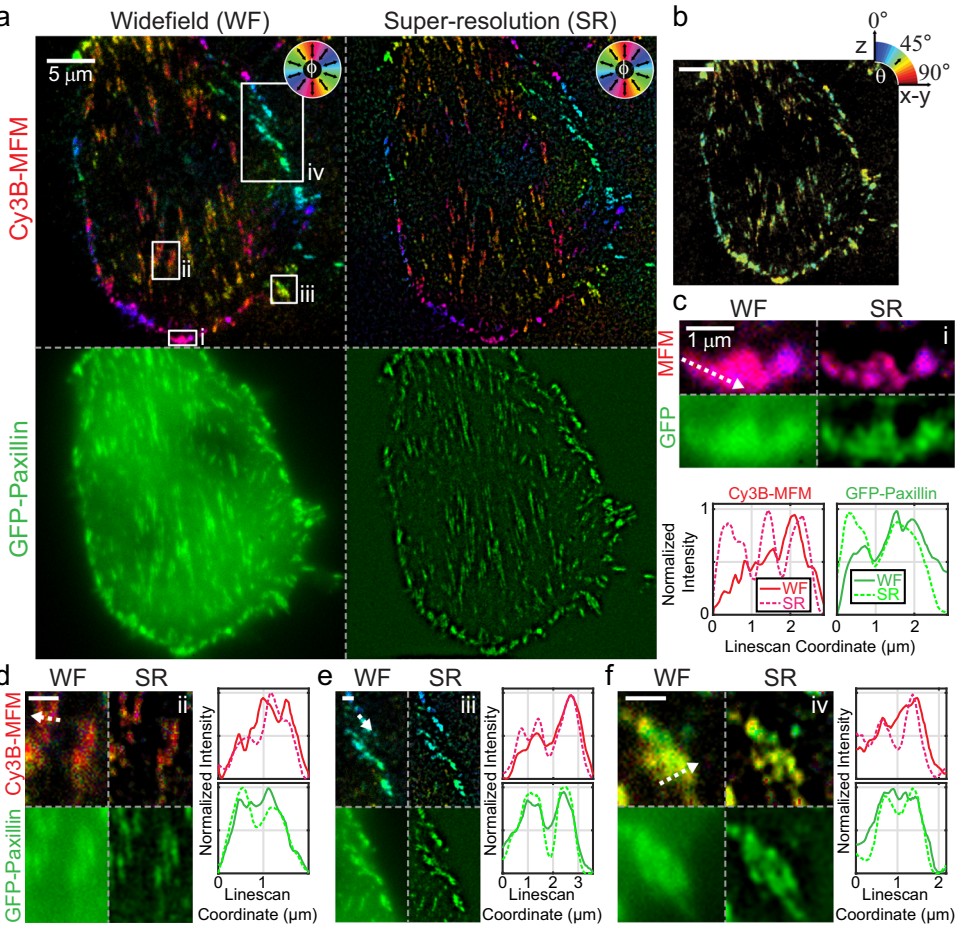

**Fig. 3 Representative 3T3 fibroblast data. a** Colormap-based display of $\phi$ values measured with SIM-MFM (top) and GFP-paxillin fluorescence (bottom) shown in both widefield (WF, left) and super-resolution (SR, bottom). Note that the $\phi$ color wheel in this subfigure is rotated by 90° relative to other color wheels in this work (this change was made to enhance the quality of display). **b** Colormap-based display of $\theta$, shown in widefield (scale bar is 5 µm). **c**–**f** Four zoom-ins (denoted in **a** with lower-case roman numerals) show FA features that can be resolved and/or sharpened using super-resolution reconstruction of SIM-MFM acquisitions (scale bars are 1 µm). Each subfigure includes four images in the same format as in **a**, along with intensity plots from a linescan denoted with a dotted arrow in the WF MFM $\phi$ map. Each plot shows the linescan of the WF image (solid line) and SR image (dotted line). In **d**–**f**, the top plot corresponds to Cy3B MFM, while the bottom plot corresponds to GFP-Paxillin. Independent experiments with 3T3 fibroblasts were repeated three times with similar results.

**Analysis of force orientation measurement error.** We were initially concerned that the use of only 3 $\alpha$ angles (compared to 73, as previously used[16]) could potentially lead to systematic errors in $\theta$ and $\phi$ measurements due to inconsistencies in the effect of measurement noise on different force orientations. To examine this effect, we performed two validations. First, we conducted Monte Carlo simulations to evaluate the effect of true $\phi$, $\theta$, and $I_{max}$ values on the accuracy (systematic and random error) of $\phi$ and $\theta$ measurements (Fig. 4a and Supplementary Figs. 10,11—see "Methods" section). Our results at the experimentally relevant signal level of $I_{max} = 1000$ photons (Supplementary Fig. 12) displayed small systematic errors in $\phi$ (less than half a degree across all orientations). As we observed in previous work[16], random error in $\phi$ was high at low $\theta$, and systematic error in $\theta$ was high at very low and high $\theta$. Importantly, we did not observe a substantial dependence in any form of error on $\phi$. While all forms of error exhibited slight periodic variations with respect to $\phi$, these variations were all smaller than 1°. The nature of these variations in error was similar across a range of $I_{max}$ from 100 photons to 10,000 photons (Supplementary Figs. 10, 11). These simulation results suggest that the use of only three $\alpha$

values introduces small $\phi$-dependent orientation measurement errors.

To test this prediction, we plotted a histogram of $\phi$ values from 81 different platelets (Fig. 4b). Because all platelets are individually organized, all $\phi$ values should be measured with equal probability, resulting in a flat histogram. However, consistent with the simulation predictions we observed slight systematic biases that resulted in a non-flat histogram with local minima close to $\phi = 17°$, 77°, and 137°. This small systematic error resulted in at most a 7% deviation from the ideal uniform distribution. These errors likely result from the small number of $\alpha$ angles (3) relative to our previous implementation of MFM (72), and in future work, it may be possible to address this issue by implementing SIM-MFM with more than 3 $\alpha$ angles. Nonetheless, we do not expect this form of systematic error, which is small in magnitude, to meaningfully bias our measurements.

As a second means of investigating potential sources of systematic error in orientation measurement, we evaluated the $\phi$ dependence of $\theta$ measurements. To accomplish this, we assembled data from several platelets and calculated $\langle\theta\rangle$ values as a function of $\phi$ (Supplementary Fig. 13). Ideally, $\langle\theta\rangle$ should not

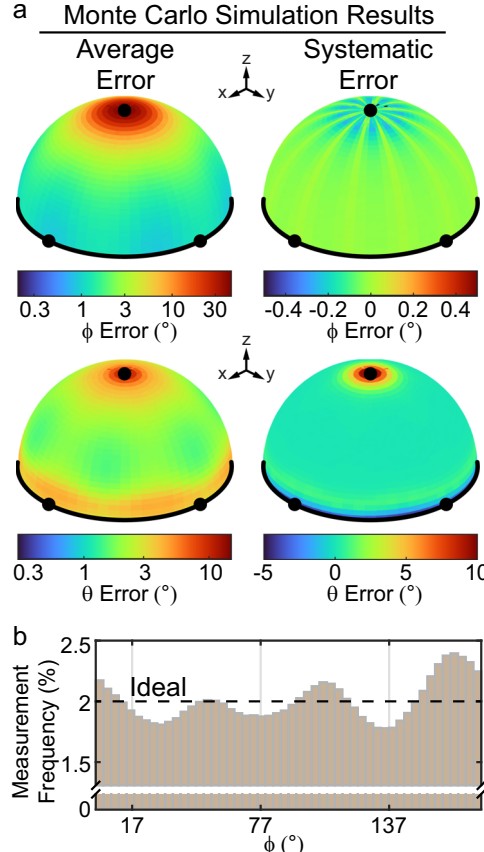

**Fig. 4 Analysis of measurement errors. a** Results of Monte Carlo simulations of force orientation measurement. Four unit hemispheres are shown. The color of each point on a hemisphere denotes the error of the orientation that passes through that point. Average error (left) and systematic error (right) are shown for $\phi$ (top) and $\theta$ (bottom). **b** Histogram of all $\phi$ values accumulated from 81 platelet timelapses. Dashed line shows the ideal behavior (2% measurement frequency in each of 50 bins). Small variations (from 1.81% to 2.16%) reveal a small effect of systematic errors. Only the brightest 50% of pixels were included in the analysis.

depend on $\phi$ in the compiled dataset. We found that $\langle\theta\rangle$ was constrained to a narrow range of $\sim 43-50°$. The small scale of this variation ($\sim 7°$) serves as a validation for SIM-MFM-based orientation measurement. However, ideally this variation would be on the order of $<1°$. This result, as well as the slight discrepancy between the $\langle\theta\rangle$ measured here and previously[16], suggest that there may be slight measurement errors that were not fully corrected for in image pre-procession (e.g., variations in the illumination profile of the microscope, or photobleaching).

**Timelapse imaging of platelets force alignment.** SIM-MFM is a versatile technique that can be implemented in dynamic, living cells without substantial loss of signal due to photobleaching. We took advantage of this feature to investigate the dynamics of platelet spreading and adhesion. Our analysis, described below, reveals that, under our experimental conditions, platelet traction force aligns over time in a manner that can only be resolved using MFM. We added platelets to the solution over a tension probe-functionalized surface and imaged several locations on the surface at 2-minute intervals for 82 min. Over the course of this acquisition, we identified 81 platelets interacting with the surface. For each of these platelets, we segmented a small region-of-interest

containing the platelet out of the imagestack (Fig. 5a). For each timepoint, we quantified the "tension area" ($T$), which is the surface area of the tension signal created by the platelet (Supplementary Fig. 14), and a measure that we call the "alignment parameter" (Fig. 5b and Supplementary Fig. 15). The alignment parameter, $R$, is a measure for the dispersion of all $\phi$ angles measured under a platelet. If all $\phi$ angles are aligned perfectly along the same axis, $R = 1$. If $\phi$ angles have an isotropic orientation, $R = 0$. In the context of directional statistics, the alignment parameter is equal to one minus the circular variance of the $\phi$ angles[72].

We found that the tension area spiked sharply, quickly plateaued within a few minutes, then gradually or sharply dropped to zero within 10–30 min. We found that the $T$ vs. time, $t$, curve for almost every platelet could be accurately fit to the equation:

$$T \approx T_{max}u(t - t_{attach})\left(\left(1 - \exp\left(-\frac{t - t_{attach}}{\tau_{spread}}\right)\right) - \left(\frac{1}{1 + \exp\left(-\frac{t - t_{attach} - t_{detach}}{\tau_{retract}}\right)}\right)\right) + T_0 \tag{9}$$

where $T_{max}$ is the maximum tension area of the platelet, $t_{attach}$ approximates the times at which the platelet attaches to the surface, $t_{detach}$ approximates the duration between platelet adhesion and detachment, $\tau_{spread}$ and $\tau_{retract}$ are time constants for platelet spreading and retraction, respectively, $T_0$ is the basal $T$ value measured (typically a noise-induced artifact) and $u(t - t_{attach})$ is the unit step function that denotes initiation of spreading at $t = t_{attach}$ (Fig. 5b, c). We constructed this equation—which is based purely off of our observation of the nature of the data—such that the first term (one mines the decaying exponential) reflects an asymptotic increase in $T$, while the second term (with the exponential in the denominator) is a sigmoid function that reflects an s-shaped decrease in $T$ to baseline. We set $T_{max}$, $t_{attach}$, $t_{detach}$, $\tau_{retract}$, and $\tau_{spread}$ all as fit parameters and used simulated annealing to fit Eq. (9) to each platelet's $T$ vs. $t$ data. We found that $T_{max} = 20 \pm 7 \ \mu m^2$ (median $\pm$ inter-quartile range), and 50 platelets had best-fit $t_{attach}$ and $t_{detach}$ values that fell within the duration of the imaging acquisition.

Of the 50 platelets for which the full process of attachment, spreading, retraction, and detachment was recorded, we quantified alignment, $R$, from $t_{attach}$ to $t_{attach} + t_{detach}$. We found that roughly a third ($N = 16$, or 32%) exhibited no significant correlation between $R$ and $t$ (Fig. 5d). Of the remainder, alignment decreased over time for 6 (12%) and 28 (56%) exhibited a gradual, asymptotic increase in alignment (Fig. 5e and Supplementary Fig. 17) that could generally be fit to the relationship:

$$R \approx R_{max} - (R_{max} - R_0)\exp\left(-\frac{t - t_{attach}}{\tau_{align}}\right) \tag{10}$$

where $R_0$ and $R_{max}$ are the minimum and maximum alignment parameters, and $\tau_{align}$ is the time constant for force alignment (only $R_{max}$, $R_0$, and $\tau_{align}$ were set as fit parameters; $t_{attach}$ from the $T$ vs. $t$ fit was used as a fixed parameter) (Fig. 5b, c). We split these cells into two groups (28 increasing-alignment cells and 22 cells with non-increasing alignment) and compared the sets of fit parameters. The dynamics of spreading were the same between the two groups ($p > 0.05$ for $T_{max}$, $T_0$, $\tau_{spread}$, $\tau_{retract}$, and $t_{detach}$ - Fig. 5f, g and Table 1). Following a thorough analysis detailed in Supplementary note 2 and Supplementary Figs. 17, 18, we eliminated the possibility that the observation of increasing alignment was a photobleaching-induced artifact.

An interesting finding of this analysis is that spreading occurs on a significantly faster timescale than alignment; $\tau_{spread} = 1.7$ min,

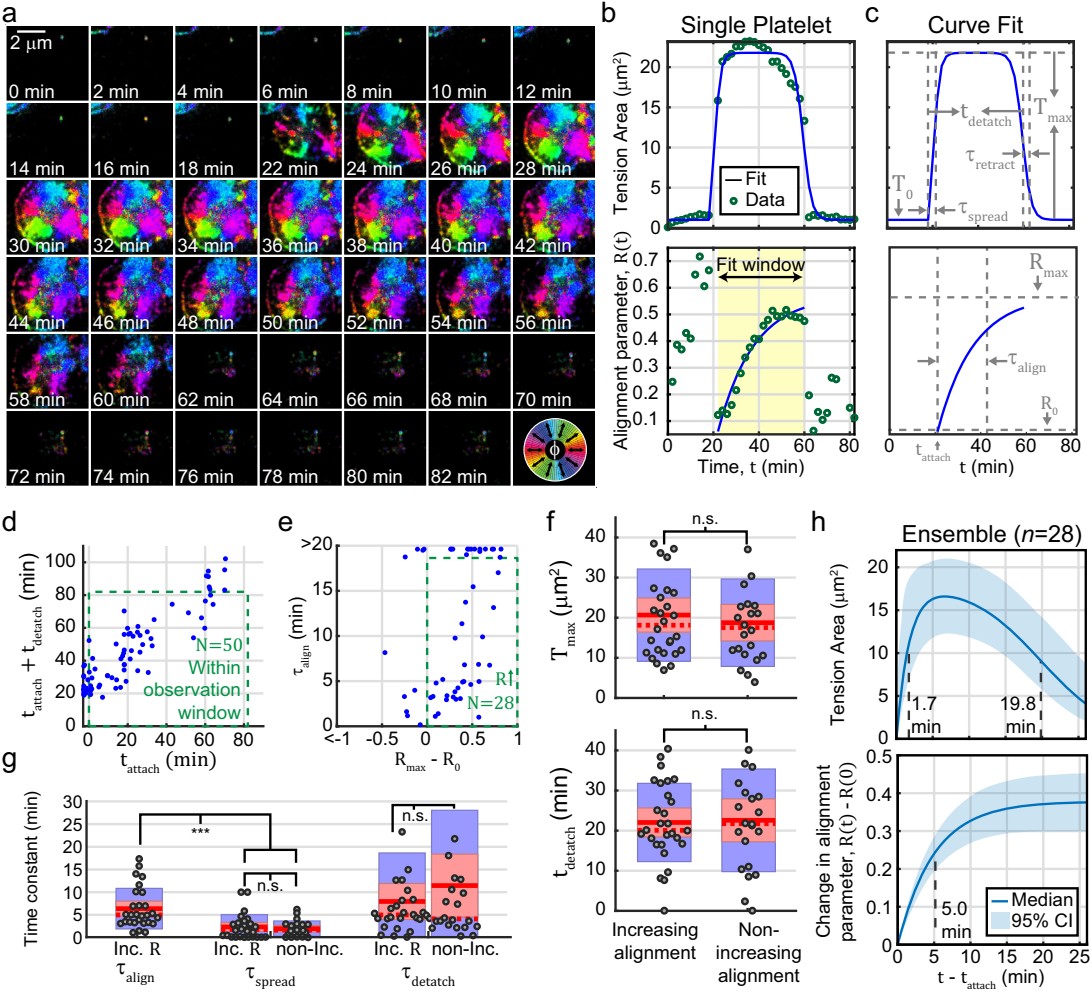

**Fig. 5 Dynamic properties of human platelets revealed using SIM-MFM. a** Timelapse of a platelet spreading on a surface, shown using the $\phi$-colormap display method (see color wheel in bottom right corner). Gradual tension alignment can be seen via careful inspection; greens and cyans are common at early timepoints, but gradually transition to purple and dark blue due to alignment of platelet tension. **b** Tension area (top), $T$, and the alignment parameter (bottom), $R$, quantified as a function of time, $t$, with curve fits shown (blue). Yellow shading denotes time window of platelet attachment. **c** Best-fit curves from **b** with annotations depicting fit parameters. **d** Scatterplot (blue dots) of best-fit $t_{attach} + t_{detach}$ (the timepoint at which platelet detachment occurs) and $t_{attach}$ (the timepoint at which attachment occurs) values. Green dashed line shows the parameter space in which attachment and detachment both occurred within the duration of the timelapse. **e** Scatterplot of best-fit $\tau_{align}$ (the time constant for tension alignment) and $\Delta R$ (the change in alignment) values for the platelets selected from **d**. Green line shows selected population with increasing alignment. **f** Boxplots of $T_{max}$ (the max platelet spread area—top) and $t_{detach}$ (the duration of platelet attachment—bottom) for the $n = 28$ increasing-alignment cells, compared with the $n = 22$ non-increasing-alignment cells. Solid red line indicates mean, dashed red line indicates median, red boxes show the standard error of the mean, and blue boxes show 95% confidence interval of the mean. Gray circles indicate individual datapoints. **g** Boxplots for $\tau_{align}$, $\tau_{spread}$ (the time constant for platelet spreading), and $\tau_{detach}$ (the time constant for platelet retraction) for the increasing and non-increasing-alignment populations. *** denotes $p = 6.9 \times 10^{-5}$, and n.s. denotes $p > 0.05$ (Wilcoxon two-sided rank-sum test for difference in population medians—see Table 1 for exact $p$-values). **h** Depiction of ensemble behavior. Blue curves denote reconstruction with median best-fit parameters and blue shading denote 95% confidence interval of the median obtained via 1,000 bootstrapping iterations with MATLAB's bootci function. Independent timelapse experiments with human platelets were repeated three times with similar results.

**Table 1 Timelapse fit parameters for platelet spreading and tension alignment: median (inter-quartile range) with two-sided Wilcoxon rank-sum test $p$-values.**

|  | $T_{max}$ (μm²) | $\tau_{spread}$ (min) | $\tau_{retract}$ (min) | $t_{detach}$ (min) | $\Delta R$ | $R_{max}$ | $\tau_{align}$ (min) |
|---|---|---|---|---|---|---|---|
| Increasing-alignment platelets (N = 28) | 18.2 (11.7, 26.6) | 1.7 (0.1, 3.1) | 5.0 (4.1, 8.3) | 19.9 (16.4, 30.0) | 0.38 (0.22, 0.54) | 0.65 (0.43,0.78) | 5.0 (3.3, 8.5) |
| Non-increasing-alignment platelets (N = 22) | 17.6 (10.9, 23.1) | 1.6 (0.6, 2.9) | 4.1 (2.4, 12.8) | 21.5 (10.9, 29.2) | 0.28 (−0.14, 0.52) | 0.61 (0.21, 0.74) | – |
| $p$-value | 0.56 | 0.93 | 0.99 | 0.79 | 0.30 | 0.28 | – |

while $\tau_{align}$ = 5.0 min ($p = 6.9 \times 10^{-5}$, Wilcoxon two-sided rank-sum test) (Fig. 5g). This result suggests that platelet activation displays a characteristic mechanical progression that includes three phases. In the first phase, the platelet spreading area and the traction forces grow until reaching a steady-state plateau. In the second phase, forces within the platelet re-organize and realign along an axis. Finally, the platelet abruptly terminates contractility and detaches from the surface. We previously only quantified tension signal but not force orientation during platelet timelapse imaging[4,73], and so the second phase appeared to exhibit a constant net traction force magnitude.

We and others previously observed alignment of platelet contractile forces along an axis (or three or more axes all pointing inward) via MFM[16], TFM[60,74], DNA tension gauge tether studies with complementary imaging of focal adhesion protein vinculin[61], and super-resolution and electron microscopy of cytoskeletal filaments and vinculin[75]. However, our work here represents, to the best of our knowledge, the first observation of alignment as a dynamic process, raising questions about the biophysical mechanisms underpinning this phenomenon. Our analysis of increasing alignment was restricted to 56% of platelets studied, as the remainder did not appear to exhibit increasing alignment. However, a localization microscopy-based analysis of human platelets spreading on fibronectin-coated glass found that platelet actin and vinculin was generally concentrated in 2, 3, or 4 lobes (65%, 19%, or 3% of platelets, respectively), while the remainder of platelet (13%) were isotropically (e.g., circularly) organized[75]. In this work, we expect that only the 2-lobe subset should be recognized as increasing alignment. This limitation occurs because, if alignment is increasing internally within 3 or 4 lobes, the whole-cell $R$ measurement should appear artificially low and remain largely time-invariant. Therefore, the non-increasing-alignment group may include 3- or 4-lobed platelets with alignment that is increasing internally within lobes. As such, we expect that our findings regarding the dynamics of alignment may applicable to greater than 56% of platelets.

Previous atomic force microscopy (AFM) or TFM-type studies of single platelets pulling on deformable substrates have shown a gradual increase in single platelet forces over tens of minutes[74,76,77], but in our MTFM work, we generally observe that the total tension signal plateaus after spreading and then eventually decreases[4,73]. The similarity between the timescale of alignment observed in this work and the timescale of contraction observed using AFM/TFM suggests that alignment may be a means through which platelets increase the efficiency of contraction; if all integrin forces are aligned along a common axis, the bulk effect on the underlying surface will be greater than if the integrins are poorly aligned—even if the summed magnitude of tension applied through all integrins is the same in both scenarios. While the potential physiological relevance and biophysical underpinnings of these findings will require further investigation, alignment of platelet tension has been shown to depend on shear flow[78], competent integrin signaling[61,75], and substrate stiffness[75]. Proper alignment was also shown to be disrupted in a patient with Glanzmann thrombasthenia[75], wherein integrin signaling is dysregulated. Together, these studies have shown that integrin signaling is an important component in the process of alignment of platelet traction and the cytoskeleton, making SIM-MFM an ideal tool for the continued study of platelet mechanics. In future work, two-color SIM-MFM may prove valuable for revealing the relationship between tension alignment and the reorganization of the tubulin and actin cytoskeletal networks. What are the roles of tubulin marginal band coiling and actomyosin contractility in controlling the dynamics of platelet mechanics? How do membrane structures

such as focal adhesions change over time and modulate traction? These questions could be answered in part using SIM-MFM in parallel with $z$-stack imaging of cell-permeable stains incubated with live platelets (e.g., Tubulin Tracker, made by Molecular Probes, Eugene, OR, USA).

Notably, a previous TFM-based study of platelet tension found that mild alignment established immediately upon platelet spreading and subsequently remained constant[74] (e.g., the first and third phases described above occur coincidentally). The discrepancy between the findings in that study and our results may have resulted from a difference in substrate stiffness (a recent super-resolution study of actin organization showed that alignment of the actin cytoskeleton along an axis was more prominent on stiffer substrates[75]), the lower spatial resolution of TFM, or differences in the identity and/or surface density of our ligands.

**SIM-MFM measurement of T-cell receptor forces.** As a final experiment in our proof-of-concept SIM-MFM study, we investigated the anisotropic mechanosensor hypothesis of T-cell receptor (TCR) activation[3]. Previous work suggested that the application of lateral tension to the TCR is more likely to stimulate an immune response compared to force that is applied perpendicular to the plasma membrane[3]. In that work, optical tweezers were used to apply tension to T cells. While this single-molecule experiment provides insights into the potential role of force orientation on TCR triggering, in physiological settings the TCR itself couples to the cytoskeleton and transmits forces that are resisted by the pMHC ligand on the apposed cell membrane. SIM-MFM offers an opportunity to measure the orientation of forces applied directly by a T-cell's own cytoskeletal activity. The observation of lateral forces applied by the TCR would support the anisotropic mechanosensor model.

We cultured primary mouse CD8$^+$ T cells on DNA hairpin tension probes that present antibodies to CD3ε (CD3ε is part of the TCR complex which includes CD3δ, γ and ζ chains, as well as the TCR α/β chains). The antibody is the clone 145-2C11, a known activating antibody that should activate T cells regardless of force orientation. A detailed discussion of our protocol for preparing surfaces functionalized with these tension probes is described in ref. [79]. As observed previously[17,79,80], T cells spread on the tension probe-functionalized surface, and the TCRs engaged and mechanically unfold tension probes to generate bright tension fluorescence signal. We imaged 26 cells and were surprised to find that the T cells did not exhibit any notable polarization-dependent response (Fig. 6 and Supplementary Fig. 19). Our results, discussed in greater detail in Supplementary Note 3, suggest that TCR force vectors lack a detectable lateral organization at the ~100 nm length scale. This is in stark contrast to traction forces generated by platelets and fibroblasts.

This result can be caused by a combination of three main factors. First, TCR forces may be perpendicular to the substrate. This is unlikely given that the T-cell membrane displays TCR-loaded projections including microvilli[81,82]. Such structures are unlikely to uniformly pull in the vertical direction. Second, tension probes may undergo fast reorientation, wherein individual open tension probes rapidly reorient within the timescale of an individual image. Third, TCR forces may be highly heterogeneous (at the sub-100 nm length scale). These last two scenarios would be surprising as frictional coupling between TCRs and actin flow is believed to generate TCR forces and actin flow is generally persistent at the micrometer length scale for timescales much longer than individual acquisitions. These results neither support nor refute the anisotropic mechanosensor hypothesis, which was tested by applying external forces to T cells rather than

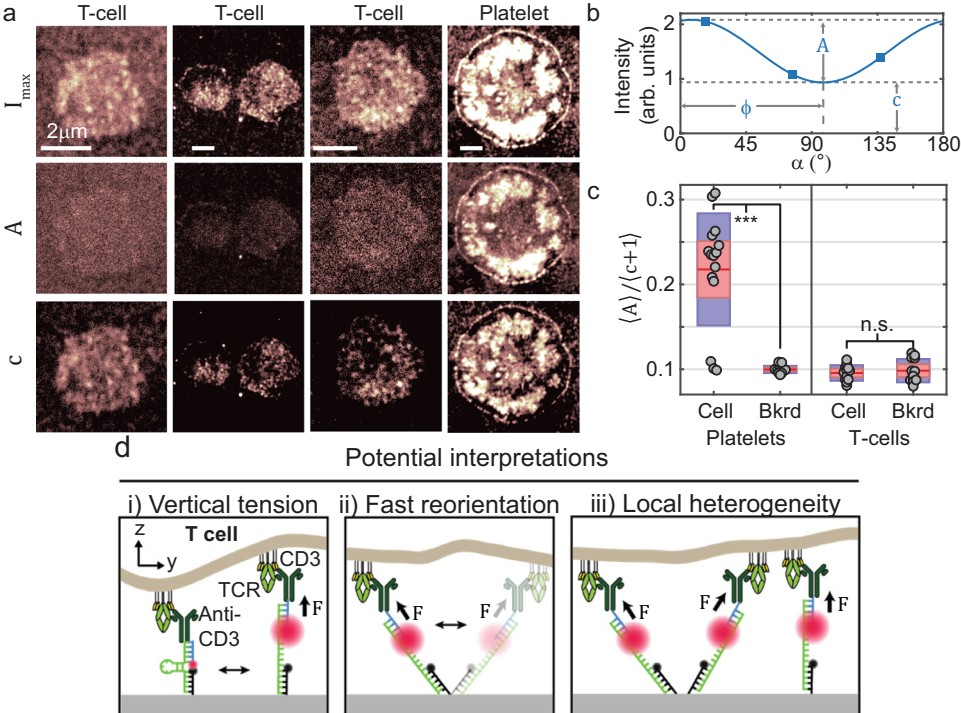

**Fig. 6 Analysis of T-cell receptor tension reveals no polarization-dependence. a** Three example T cells and one platelet are shown in three panels each. The top row shows $I_{max}$, the second row shows the polarization-dependent component of the fit sinusoid, $A$, and the third row shows the polarization-independent component of the fit sinusoid, $c$. While platelets display substantial polarization-dependence (high $A$), T cells exhibit almost no polarization-dependence (low $A$). **b** A fit sinusoid is shown with $A$ and $c$ depicted (adapted from Fig. 1c). **c** A boxplot-based quantification of the polarization response, $\langle A \rangle/(c+1)$ (the averaged polarization-dependent component divided by the average polarization-independent component of the fit sinusoids) of $n = 25$ T cells and $n = 17$ platelets, both under the cell and in the surrounding background close to the cell. A more detailed description of this analysis can be found in Supplementary note 3. Solid red line indicates mean, dashed red line indicates median, red boxes show the standard error of the mean, and blue boxes show 95% confidence interval of the mean. Gray circles indicate individual datapoints. While platelets exhibit a much higher polarization response than the background signal (*** denotes $p = 3.7 \times 10^{-5}$, Wilcoxon two-sided rank-sum test), T cells do not (n.s. denotes $p = 0.93$). The images shown here were collected during a single experiment, but unpublished experiments with T cells were previously repeated three times (using a previous iteration of MFM[16]) with similar results. **d** Three potential causes of the lack of polarization response in T cells. TCR forces may be some combination of (i) vertical to the glass surface, (ii) highly dynamic such that the orientation of each open tension probe is randomized during the ~100 ms exposure time of each image, and (iii) locally heterogeneous such that neighboring probes obfuscate each others' polarization responses. All three of these scenarios can result in the lack of polarization response observed.

measuring their intrinsically applied forces[3]. Instead, our findings show that if lateral forces are indeed a potent trigger signal for TCR activation, then these forces will be difficult to image as they are likely to be disorganized and/or highly transient. Accordingly, there is a need for further development in this area to simultaneously capture TCR force orientation and TCR triggering with a molecular resolution, potentially by combining SIM-MFM with techniques for single-molecule visualization of cell-generated forces[18,48].

## Discussion

We have demonstrated that a SIM microscope can be used to implement MFM for rapid mapping of molecular force orientation. The inherent super-resolution capability of SIM allows the mapping of force magnitude information with ~100 nm resolution. To demonstrate the broad applicability of this technique, we used SIM-MFM to image platelets, fibroblasts, and T cells. We have shown that the resolution enhancement afforded by SIM can reveal traction structures that are not resolvable with conventional widefield imaging methods. While this resolution enhancement could in theory be accomplished by applying computational approaches to conventional MFM data sets[83], the

SIM-MFM approach has the added benefit of enabling MFM using commercially available SIM microscopes in a turn-key fashion. SIM-MFM also enables parallel rapid super-resolution imaging of structural proteins such as paxillin and facile, photostable timelapse imaging of molecular force orientation. In the future, it may be possible to perform two-color SIM-MFM using probes with other duplex terminus-stacking dyes such as cyanine 5 (Cy5). Such an advance could enable simultaneous mapping of forces applied through probes that bind to different surfaces receptors, or probes that exhibited different $F_{1/2}$ values. We expect that the ongoing development of the SIM and pSIM techniques will continue to improve SIM-MFM acquisition[84–86].

The moderate signal-to-noise ratio (SNR) is a key factor limiting super-resolution reconstruction of SIM-MFM images. The primary factor constraining SNR is photobleaching of the tension probes' fluorophores; the need for several high-quality images per SIM acquisition (and the need for several acquisitions during timelapse imaging), means that moderate laser powers and short exposure times must be used to ensure that substantial photobleaching does not occur between images. Accordingly, techniques that improve photostability (e.g., the incorporation of DNA fluorocubes[87] and/or the use of in-solution oxygen scavenging systems[88]) could enable higher SNR, longer imaging durations,

and higher frame rates. A broader limitation of pSIM-based techniques such as SIM-MFM is that super-resolution reconstruction only improves the resolution of the intensity map – not the force orientation map. This limitation arises because the orientation of the polarization and the illumination striping are rotated simultaneously, rather than separately. In other words, the spatio-angular cross-harmonics cannot be detected with conventional SIM microscopes[53]. While it may be possible to engineer optical systems that can resolve cross-harmonics by modulating the polarization and diffraction patterns separately, it is unlikely that such acquisitions could be implemented using existing SIM microscope hardware. Finally, a key limitation of MFM is that force orientation measurement is degenerate (i.e., each measured orientation could be one of two orientations that mirror each other across the $z$ axis). Perhaps eventually, SIM will be performed with $E$ dipoles that are not parallel to the coverslip, which would enable nondegenerate force vector mapping[41]. Again, such advances will likely not be possible without hardware modifications to existing SIM microscopes.

In our proof-of-concept studies, we have demonstrated that platelets undergo traction force realignment on a timescale that is substantially longer than the timescale of platelet spreading (5 vs. 1.7 min, respectively). This finding raises important questions about the dynamic processes governing platelet mechanics. Future work could use timelapse SIM-MFM, along with two-color imaging of structural proteins such as tubulin, actin, and vinculin[75], to investigate relationships between cytoskeletal network re-arrangement, focal adhesion growth, and tension alignment. Moreover, SIM-MFM could be a powerful tool for studying platelet-related diseases such as Glanzmann thrombasthenia or Wiskott–Aldrich syndrome (in which actomyosin contractility is impeded)[89]. It may also be necessary to investigate the dynamic effects of platelet biochemical activity on tension probe-functionalized surfaces; it was previously shown that platelets secrete ECM proteins such as fibronectin while spreading in vitro[90], which can potentiate $\alpha IIb\beta 3$ mediated adhesion. Because cRGDfK is specific to $\alpha v\beta 3$ and $\alpha 5\beta 1$, the gradual potentiation of $\alpha IIb\beta 3$ signaling (driven by the accumulation of ECM proteins secreted onto the surface surrounding/below platelets) could introduce temporal effects on platelet mechanics. Finally, SIM-MFM is well-suited to investigating the effects of physiologically relevant shear flow rates on platelet mechanics and dynamics[78].

We have also shown that TCR forces are, in contrast to the integrin traction forces that we generally study with MFM, not temporally stable, homogeneously aligned, and lateral. We expect that SIM-MFM will be used in several future applications in cellular biology and for the investigation of force generation by active nanomaterials, which is an emerging application area of DNA mechanotechnology tension probes[91,92].

## Methods

**Materials**. Unless otherwise stated, all materials were purchased from Sigma-Aldrich or ThermoFisher and used without additional purification.

**Preparation of supported lipid bilayer (SLB) functionalized microparticles**. SLB-functionalized microparticles were prepared as previously described[16]. Briefly, small unilamellar vesicles (SUVs) with (average diameter of 100 nm) were prepared via lipid extrusion; 4 mg mL$^{-1}$ of 1,2-dioleoyl-$sn$-glycero-3-phosphocholine lipid (DOPC) (Avanti Polar Lipids Inc., Alabaster, Alabama) was diluted in ~500 μL chloroform in a round-bottom flask. The chloroform was removed via rotary evaporation, resulting in the formation of a thin lipid film. The lipid film was then dried under a steam of N$_2$ and subsequently hydrated with 3 mL of milli-Q water and sonicated prior to three freeze-thaw cycles to reduce multi-lamellarity. The lipid solution was then extruded 10 times through an 80 nm polycarbonate filter using 10 mL LIPEX extruder (Northern Lipids/TRANSFERRA Nanosciences Inc.). Extruded SUVs were clearer and less translucent than their precursors, suggesting unilamellar structure. SUVs were stored at 4 °C and then used within 4–6 weeks.

To prepare SLB-coated microparticles, 100 μL 1 mg mL$^{-1}$ 5 μm silica beads (product #: SS06N, Bangs Laboratories, Fishers, Indiana) were mixed with 100 μL of SUVs and equilibrated on a rocker for 15 min at room temperature. The supported lipid bilayer (SLB) beads were purified via three successive 5 min spin in a benchtop centrifuge at 2000 r.p.m. After each spin, the supernatant was removed via pipette aspiration and was replaced with 1 mL of 1× PBS. The SLB-functionalized beads were then incubated with 5 μM DiI (product #: CN-1006 ThermoFisher Scientific, Waltham, Massachusetts) for 15 min. The DiI-loaded SLB-functionalized beads were separated from free DiI with three successive 5 min spins, with each spin followed by supernatant aspiration and replacement with 1× PBS as before. The final volume was 100 μL. The mixture was vortexed, and 2 μL of SLB-functionalized beads were then diluted in 1 mL of 1× PBS to ensure that particles would be spatially isolated. The diluted beads were then vortexed and 100 μL were transferred to one clean well of a flow chamber (product #: μ-Slide VI 0.4, Ibidi, Verona, Wisconsin).

**Isolation and handling of platelets**. Experiments were performed with platelets drawn from human volunteers. All procedures using donor-derived human platelets were approved by the Institutional Review Board of Children's Healthcare of Atlanta/Emory University (IRB # IRB00006228). Written, informed consent was received from participants prior to their inclusion in studies. Venous blood was drawn from the arm of human volunteers in two 3 mL portions, the first portion discarded. Next, 0.75 mL of Anticoagulant citrate dextrose (ACD) solution (0.75 mL) was added to the second portion. The mixture of whole blood and ACD was then spun at 140 RCF for 12 min. The lowest centrifuge acceleration and braking settings were used to avoid platelet activation. The platelet-rich plasma was collected, combined with 10% ACD (v/v) and 3 μM Prostaglandin E1, and spun at 900 RCF for 5 min with the lowest centrifuge acceleration and braking settings. The platelet-poor plasma was aspirated following the spin. Then, the platelets were suspended gently in 2 mL 1× Tyrode's buffer with 0.1% BSA w/v. Finally, platelets were allowed to settle for ~1 h at room temperature before the start of each experiment, which allowed them to return to a resting state. Platelets were used within 8 h. To add platelets to surfaces, the platelets were gently inverted multiple times to promote dispersion and were added to the imaging chamber, which was loaded with Tyrode's buffer, in 10–20 μL increments until the desired surface density was reached. Experiments with platelets were repeated five times using different donors and surface preparations to ensure reproducibility of the technique. Platelets were imaged between 10 min and 2 h after plating.

**Culture and handling of 3T3 fibroblast cells**. NIH 3T3 cells were cultured to 75% confluency in Dulbecco modified essential medium (DMEM) supplemented with 10% (v/v) fetal bovine serum and 2.1 mM L-Glutamine, 1000 μg L$^{-1}$ streptomycin, and 10,000 IU L$^{-1}$ penicillin G. Cells used in different experiments were derived from different subcultures. Cells were removed from culture flasks and replated to DNA hairpin-functionalized surfaces within 20 min. For removal from flasks, cells were removed using a trypsin solution (Corning) containing 0.25% (wt/vol) trypsin, sodium bicarbonate, and 2.21 mM EDTA. The cells were then centrifuged at 218 rcf, pelleted, and resuspended in the same medium. Next, the cells were transferred to and plated on DNA hairpin-functionalized surfaces at a density of 20,000 cells cm$^{-2}$. Cells were imaged between 10 and 60 min after plating. Experiments with 3T3 fibroblasts were repeated three times using different surface preparations to ensure reproducibility of the technique.

**Surface preparation**. Glass coverslips with thicknesses of 0.17 mm—either 25 mm × 75 mm rectangular coverslips (product #: 10812, Ibidi, Verona, Wisconsin) or 25 mm diameter round coverslips (product #: 48382-085, VWR, Radnor, Pennsylvania)—were cleaned via sonication in ethanol for 10 min and then dried in an oven at a temperature of 100 °C for 20 min. The dried coverslips were then exposed in piranha acid (a solution of 30% (v/v) H$_2$O$_2$ in concentrated H2SO4) (Caution! This solution is extremely corrosive and explosive in contact with organics!) for 10 min. Coverslips were then rinsed six times with water and then rinsed three times with ethanol. The rinsed coverslips were then functionalized with amine groups. This was achieved by incubating slides in 3% (v/v) (3-aminopropyl)triethoxysilane in ethanol for one hour. Next, the slides were washed three times with ethanol, baked in an oven at 100 °C for 20 min, and then washed 3 times with ethanol. The slides were then baked in a 100 °C oven (again) for 20 min. A solution of 2 mg mL$^{-1}$ Biotin-NHS in anhydrous DMSO was then placed onto the dried slides, which were then sealed with parafilm in plastic Petri dishes and stored at room temperature for 12 h or longer. These surfaces were then used within two weeks of preparation. On the day of use, the biotinylated coverslips were sonicated in ethanol for 15 s, rinsed with ethanol, and dried with N$_2$. Round slides were then mounted into round imaging chambers (product #: A7816, ThermoFisher, Waltham, Massachusetts) and immediately immersed in 1× PBS. Rectangular slides were attached to Ibidi sticky slides (product #: 80606, Ibidi, Verona, Wisconsin), and each of the six wells was immediately immersed in 1× PBS. Next, the mounted surfaces were rinsed 5 times with 1× PBS, passivated for 45 min with a solution of 1 mg mL$^{-1}$ BSA in 1× PBS, washed again with 1× PBS, incubated in 0.05 mg mL$^{-1}$ streptavidin in 1× PBS for 1 h, and washed once more with 1× PBS. Finally, the streptavidin-functionalized surfaces were incubated with

20 nM hybridized DNA hairpin probes in 1× PBS for 1 h, washed again with 1× PBS, and washed with 3 volumes of imaging buffer. Cells were then plated onto the surfaces for imaging.

**Synthesis of cRGD-presenting strand**. The cRGDfK-presenting strand was synthesized as described previously[18]. To generate generate cRGD-azide 100 μgof cyclic-arginine-dlicine-aspartic acid-D-phenylalanine-lysine(PEG-PEG) (cRGD) (Vivitide product PCI-3696-PI) with 2 mg NHS-Azide(Thermo product 88902) and the product cRGD-azide purified by RP-HPLC with a flow rate of 1 ml/min with a linear gradient of 10–60% B over 50min. The product cRGD-azide was then dried and mixed with 5 μlof 1 mM DNA cRGDfK-presenting strand (Table S1) purchased from Integrated DNA technologies and 1 μl of 10 X PBS pH 7.4. Then 2 μl of 50 mM tris(benzyltriazolylmethyl)amine was first combined with 1 μl of 20 mM copper(II) sulfate, then 1 μl of 20 mM sodium ascorbate was added to this mixture.The mixture was then added to the cRGDfK-presenting strand solution in a microcentrifuge tube, covered with a blanket of nitrogen gas for 10 seconds to reduce oxygen concentration, sealed with parafilm, and sonicated for 2 hrs. The product of the copper catalyzed click reaction between cRGD-azide and the alkyne substitution on the cRGDfK-presenting strand was then purified using RP-HPLC as before. The product cRGDfK-presenting strand-cRGD was then dried and combined with 50 μg of Cy3B-NHS dissolved in 1 μl of anhydrous dimethylsulf-oxide, 1 μl of 10 X PBS pH 7.4, 1 μl of 1 M sodium bicarbonate, and 7 μl of nanopure water and sonicated for 2 hrs. The product of the acylation of the amine substitution on the cRGDfK-presenting strand with Cy3B-NHS was then purified by RP-HPLC, then dried and dissolved in nanopure water and stored at −20 °C until use.

**DNA strand hybridization**. Modified DNA oligonucleotides were hybridized together at concentrations of 200 nM (for the hairpin-strand) and 220 nM (for two handle strands) in 1× PBS, representing a molar ratio of 1.1:1.0:1.1 anchor-strand/hairpin-strand/ligand-strand. Strand sequences, which were purchased from Integrated DNA technologies (Coralville, IA), are shown in Supplementary Table 1. Annealing was carried out by heating to 95 °C, then cooling to 25 °C at a rate of 1.3 °C min⁻¹ in a Perkin-Elmer GeneAmp 2400 ® thermocycler. Annealed probes were diluted in 1× PBS before being added to streptavidin-functionalized surfaces.

**Structured illumination microscopy (SIM)**. SIM images were acquired on a Nikon N-SIM system equipped with a CFI Apo ×100 1.49 NA objective, an Andor iXon EMCCD (60 nm per pixel—DU-897, Andor Technology, UK), and 488 and 532 nm solid-state lasers. Cells were generally identified using low laser illumination power to avoid photobleaching. Before each image, the focus was manually adjusted until the stripe pattern of the illumination profile could be clearly observed in the fluorescent background. For each N-SIM image, fifteen or nine images of a sample were acquired in different phases (via the built-in 3D SIM and 2D SIM modes, respectively). For timelapse imaging, the Nikon Elements software "ND acquisition" feature was used to save several platelet-containing locations on the surface and then initiate a long multipoint timelapse. We found that, while there was some *xy*-drift, the *z*-focus of all locations was generally preserved for the full duration of these acquisitions. We generally used an exposure time of 100 ms, 4.9× conversion gain, and a gain multiplier of 100.

To confirm linear polarization of the excitation laser, a linear optical polarization filter (WP25M-VIS, Thorlabs, NJ) on a rotatable mount (RSP1/M, Thorlabs, NJ) was placed between the objective and an optical power meter (7Z02621, Ophir Starlite, Ophir Photonics, PN) and set to 534 nm detection mode. The polarization axis of the polarization filter was aligned normal to the polarization axis of light exiting the objective at 0° by rotating the mount until the reading by the power meter was minimized. The 532 nm laser was set to operate at 100% of its full power in SIM mode. The polarization filter was then rotated in 20° increments, and the power of the light exiting the objective was recorded at each 20° increment. Three replicates of this experiment showed that the laser intensity varied sinusoidally with respect to polarizer angle (Supplementary Fig. 1). The intensity minimized at zero, thus confirming linear polarization of the laser.

**Standard resolution image processing**. Raw.nd2 Files from SIM acquisitions were opened in MATLAB 2018b using the bfopen MATLAB function (Bioformats package[93]). Raw images were originally in a 3 × 5 montage form, wherein the 15 images that make up a single acquisition are concatenated directly together. Reconstruction was only performed on 3D SIM acquisitions, which exhibited clearer striping patterns than 2D SIM on our microscope. Each raw montage was loaded into MATLAB, and the CCD background of 200 intensity units was sub-tracted from it. The background-subtracted montage was then divided by a background correction montage to account for intensity variations between and within individual images (see below). Finally, the 1st percentile of the montage (calculated using MATLAB's quantile function) was subtracted from the entire montage, and all negative values were set to zero. (This previous step was skipped when processing timelapses and during the photobleaching analysis discussed in Supplementary Figs. 17 and 18). Next, the montage was split into 15 individual images. Each set of five images acquired at a common polarization angle was then averaged, resulting in three images—one corresponding to each of the three

polarization angles ($\alpha_1$, $\alpha_2$, and $\alpha_3$). These three images were then used to calculate $I_{max}$, $\phi$, and $\theta$ using Eqs. (4–8).

**Preparation of illumination profile correction images**. Background correction images were generated on the day of the experiment by collecting at least six SIM acquisitions with the microscope focused onto the bottom of a Chroma slide or a surface with 100% open tension probes, which were prepared by adding a "opening strand" (which is complementary to the full hairpin) during the DNA strand hybridization step. Each acquisition produced a 3 × 5 montage of images. An analysis suggested that the two methods show highly similar background illumi-nation profiles (Supplementary Fig. 5). All montages were opened into MATLAB, baseline (200 arb. units) subtracted, and smoothed with a rolling ball filter (using MATLAB's imfilter command) with a 3-pixel radius. The filtered montages were then averaged together. Finally, the montage was normalized to the mean value of the entire montage. This process was repeated each time an experiment was performed.

**Super-resolution image processing**. Super-resolution image reconstruction was performed using built-in features of the Nikon Elements software on the computer used to control the SIM microscope. Reconstructions were previewed using the "thumbnail" feature and reconstruction parameters (illumination contrast, noise suppression, and out-of-focus blur suppression) were adjusted manually to max-imize the apparent spatial resolution while minimizing reconstruction artifacts (honeycomb, dotting, etc.). All platelet images were reconstructed in the "batch reconstruction" mode with reconstruction parameters of illumination contrast = 1.45, noise suppression = 1.44, out-of-focus blur suppression = 0.18. The fibroblast image shown in Fig. 3, was reconstructed with parameters of 0.44, 0.10, and 0.19, respectively. The fibroblast image shown in Fig. 3, was reconstructed with para-meters of 0.44, 0.10, and 0.19, respectively. The reconstruction parameter sets for the fibroblasts shown in Supplementary Fig. 8a–c were [1.45, 1.44, and 0.18], [3.27, 1.50, and 0.19], and [1.22, 2.39, and 0.16], respectively.

**Error simulations**. We adapted a Monte Carlo simulation process described in ref. [41]. in MATLAB 2019b. In these simulations, we generate ideal data for a given ($\phi$, $\theta$, $I_{max}$) combination using Eqs. (2) and (3) in the main text, add in noise, and then calculate $\phi^*$, $\theta^*$, and $I^*_{max}$ (where * denotes the value measured) using Eqs. (4–8). We repeat this process ~30,000 times, using randomly generated noise values each time. We then quantified systematic error ($\sigma$) and random error ($\varepsilon$) for $\theta$ and $\phi$, as follows:

$$\varepsilon_\phi = \left\langle |\phi - \phi^*| \right\rangle \tag{11}$$

$$\sigma_\phi = \phi - \left\langle \phi^* \right\rangle \tag{12}$$

$$\varepsilon_\theta = \left\langle \left| \theta - \theta^* \right| \right\rangle \tag{13}$$

$$\sigma_\theta = \theta - \left\langle \theta^* \right\rangle \tag{14}$$

This process is described in greater detail in the online methods section in ref. [41].

**Timelapse processing**. Each timelapse was processed to account for stage drift and photobleaching using an automated MATLAB code. First, each image in each timelapse was processed as described in "Standard resolution image processing" (except for the last sentence, which describes calculating force orientation). Second, drift correction was performed by using MATLAB's built-in "imregtform" com-mand with the default multimodal settings to measure drift between successive pairs of images. Following each calculation, the later image in the pair was then translated using the built-in MATLAB function "imwarp", ultimately resulting in a timelapse with very small amounts of drift. Next, a large, platelet-free rectangular region of each timelapse was manually selected. The average fluorescence intensity within this region was then calculated for each timepoint. The background intensity vs. time curve was then fit to an exponential decay function. Finally, the entire timelapse was normalized to the exponential decay function, which largely corrects for photobleaching. Finally, the drift- and photobleaching-corrected timelapses were used to calculate $\phi$ and $I_{max}$. Additional post-processing of time-lapse data was performed using methods described in Supplementary Figs. 14, 15, 17, and 18.

**Primary T-cell isolation and use**. All animals were housed and bred at the Division of Animal Resources at Emory University, in accordance with the Insti-tutional Animal Care and Use Committee (IACUC) protocol (PROTO201800239) of the IACUC of Emory University. Briefly, OT1 mice were sacrificed and the spleen was harvested for T-cell isolation. OT1 T cells were purified using a Miltenyi Biotec CD8a⁺ T-cell isolation kit (130-104-075) following their standard protocol. Isolated T cells were stored on ice in Hanks' Balanced Salt Solution (Millipore Sigma, H8264) until use. T cells were seeded on surfaces at a density of 200,000 cells mL⁻¹ in Hanks' Balanced Salt Solution and allowed to spread on the surface for 10 min prior to image acquisition. All T-cell data presented in this work was

obtained from T cells collected from a single mouse. However, T cells were imaged on three independent surfaces, and a platelet experiment was performed on the same day to confirm the proper alignment of the microscope system. In addition, numerous unpublished experiments performed using the original MFM technique yielded the same polarization-independence observed in this work.

**Gold nanoparticle surface preparation**. Gold nanoparticle surfaces for TCR tension experiments were functionalized as described previously[17,80]. Briefly, No. 2 glass coverslips were sonicated with nanopure water for 5 min and then sonicated with 200 proof ethanol for 5 min in a Teflon rack. Slides were then placed in an 80 °C oven until all ethanol had evaporated. Slides were submerged in a fresh piranha solution (3:1 mixture of $H_2SO_4$ and $H_2O_2$, caution: extremely hot and highly explosive if mixed with organics) for 30 min. After 30 min, slides were transferred to a beaker containing 40 mL nanopure water and rinsed 6 times. Slides were then rinsed 3 times in 200 proof ethanol to remove all water and incubated in a 3% APTES (Millipore Sigma, 440140) solution in 40 mL of 200 proof ethanol for 1 h at room temperature. Surfaces were then washed 3 times in 200 proof ethanol and cured in an 80 °C oven for 30 min. The now amine-modified surfaces were placed in a petri dish lined in parafilm and coated with 200 μL of 0.5% w/v lipoic acid-PEG-NHS (Biochempeg, HE039023-3.4 K) and 2.5% w/v mPEG-NHS (Biochempeg, MF001023-2K) in 0.1 M $NaHCO_3$ for 1 h. The PEG solution was rinsed off of slides with nanopure water, and slides were placed back in Petri dishes and coated with 1 mg mL$^{-1}$ of sulfo-NHS-acetate (ThermoFisher Scientific, 26777) in 0.1 M $NaHCO_3$ for 30 min. Surfaces were rinsed with nanopure water and incubated with a 20 nM solution of gold nanoparticles (Nanocomposix, custom synthesized, 8.8 nm diameter, tannic acid modified) for 30 min. During this incubation, 4.7 pN hairpin, Cy3B, and BHQ2 strands that form the DNA hairpin probe construct were mixed and annealed at a ratio of 1.1:1:1 in 1 M NaCl at a 300 nM concentration. After annealing, an additional 2.7 μM of the BHQ2 strand was added to the DNA solution for better quenching. The gold nanoparticle solution was rinsed off of slides using nanopure water after incubation. Then, 100 μL of the annealed DNA solution was sandwiched between two slides and incubated overnight at 4 °C. The next day, slides were washed in 1× PBS and 64 μg mL$^{-1}$ of streptavidin (Rockland, S000-01) in PBS was added to each slide for 1 h. Slides were washed again with 1× PBS and 40 μg mL$^{-1}$ of biotinylated anti-CD3ε (BioLegend, 100304) was added to slides for 1 h. Slides were rinsed with 1× PBS and mounted in imaging chambers. Hanks' Balanced Salt Solution was immediately added to the imaging chambers to prevent slides from drying. Surface quality was confirmed by fluorescence intensity and then used for experiments immediately after.

**Reporting summary**. Further information on research design is available in the Nature Research Reporting Summary linked to this article.

## Data availability

Source data are provided with this paper[94]. Raw SIM acquisitions, with corresponding background illumination correction images, of platelets (Figs. 1, 2–6 and Supplementary Figs. 3–7, 9, 12, 13, 19), 3T3 Fibroblasts (Fig. 3 and Supplementary Figs. 8, 9), T cells (Fig. 6 and Supplementary Fig. 19), as well as platelet timelapses (Fig. 5 and Supplementary Figs. 14, 16–18), are freely available at https://doi.org/10.15139/S3/FXOHWV. Additional data sets generated during the current study (i.e., processed data files generated by MATLAB code) are available from the corresponding author on reasonable request. Representative raw data from a SIM-MFM acquisition, as well as raw data used for illumination profile correction, are included with the supplemental software available at GitHub (https://github.com/ABlanc9/SIM_MFM).

## Code availability

Software used for single-acquisition reconstruction, along with representative data, can be downloaded from GitHub (https://github.com/ABlanc9/SIM_MFM). Additional MATLAB scripts are available from the corresponding author on reasonable request.

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

## Acknowledgements

J.M.B., A.T.B. and R.G. acknowledge NSF GRFP grant no. 1444932. J.M.B. acknowledges NCI fellowship grant no. F99CA234959. A.T.B. acknowledges NCI fellowship grant no. F99CA245789. A.V.K. acknowledges NIH grant no. F31 F31CA243502. M.E.Q. acknowledges NIH grant no. F31 F31HL134241. R.L. acknowledges NIH grant no. HL082808. A.L.

M. and K.S. acknowledge NIH grant no. R01GM131099. A.L.M. acknowledges NSF CAREER1832100. K.S. acknowledges NIH grant no. R01GM124472. We thank Laura Fox-Goharioon, Neil Anthony, and William Giang of the Integrated Cellular Imaging Core Facility at Emory University for maintaining and troubleshooting the SIM microscope used in this work. We also thank Eric Rentchler of the Biomedical Microscopy Core at the University of Michigan for coordinating access to software for SIM super-resolution reconstructions. We thank Rong Ma of Emory University for providing graphical illustrations. Finally, we thank Tejeshwar Rao, Reena Beggs, Tomasz Nawara, and Will Dean of the Mattheyses lab at the University of Alabama, Birmingham for helpful conversations.

## Author contributions

A.T.B., J.M.B., A.L.M., and K.S. conceived the project. A.T.B. performed experiments and simulations and analyzed data. J.D.C. assisted with all experiments and surface preparation. A.V.K. assisted with cell handling and surface preparation for T-cell experiments. R.G. and A.S.B. assisted with supported lipid bilayer experiments and instrument calibration. W.C., M.E.Q., and R.L. assisted with platelet handling. J.M.B. and H.S. assisted with platelet and fibroblast experiments. R.L.B. assisted with fibroblast handling. The manuscript was written with input from all authors.

## Competing interests

The authors declare no competing interests.
