## [Peer Review File · Nature Communications]

Reviewers' Comments:

Reviewer #1:

Remarks to the Author:

The manuscript by Blanchard et al. is a systematically designed, meticulously performed, and well-written proof-of-concept study on the feasibility of 3D-SIM in combination with molecular force microscopy to assess forces generated at receptor-ligand interfaces and importantly nanoscale organization and orientation of these forces based on transition dipole moments of fluorophore-conjugated to a DNA tension probe. In my opinion, the entire study documented in this paper of great interest not only to the light microscopy imaging community but across broad interdisciplinary fields spanning cell biology, mechanobiology/mechanochemistry, and biophysics. Blanchard et al. using SIM-MFM, provide biologically relevant key takeaway messages with their studies on human blood platelets and primary T-cells from mouse and in-vitro cell line such as the 3T3 fibroblast, supported by solid datasets and detailed analysis. In addition, the authors also provide extensive details on the implementation of SIM-MFM for potential future users.

In summary, the current version of the manuscript needs no additional experiments to substantiate the findings as observations and the interpretations thereof are well within the scope of the manuscript.

While I am limiting myself to the platelet part of the manuscript, my specific queries to the authors are as follows:

1. Imaging protocols, parameters, and data analysis: I would highly recommend the authors to prepare a step-by-step protocol with a visual guide on the implementation of SIM-MFM on the Nikon N-SIM system for image acquisition and post-acquisition image processing steps and publish this into a citable repository such as the Nature Protocol Exchange. In my opinion, these protocols form the core of this paper and hence will be relevant for users who intend to start testing SIM-MFM on their own.

2. Can the authors comment on the influence of changes in the local microenvironment at the ligand-receptor interface on the plasma membrane especially in platelets that actively secrete their acidic granular content during spreading on the transition dipole moment for Cy3b and how it may influence ϕ , θ and I_{max} ? Are there other fluorophores that show similar transition dipole moment as Cy3b? If yes, is it possible to combine two or more DNA tension probes with unique ligands to target different receptors within the same experiment? or are there limitations? Please discuss.

3. In the current study authors used cRGDfK as a high-affinity ligand for $\alpha v\beta 3$ and $\alpha 5\beta 1$ integrin which also bind to fibronectin with high affinity. Previously it has been shown that platelets secrete or "self-deposit" their own ECM proteins such as fibronectin (and maybe other ligands) while spreading in Vitro and this potentiates $\alpha I I b \beta 3$ mediated platelet adhesion/spreading beyond the available ligand covered surface. (PMID: 25964667). Can the authors provide additional comments on how this phenomenon may influence the current observations on the observed differences in force tension points and their orientation along lamellipodia and in the inner regions of a spreading platelet during the course of timelapse measurements?

4. To quote the authors "An interesting finding of this analysis is that spreading occurs on a significantly faster timescale than alignment; $\tau_{spread}=1.7$ min, while $\tau_{align}=5.0$ min ($p<0.001$, Wilcoxon rank-sum test) (Fig. 5g). This result suggests that platelet activation displays a characteristic mechanical progression that includes three phases. In the first phase, the platelet spreading area and the traction forces grow until reaching a steady-state plateau. In the second phase, forces within the platelet re-organize and realign along an axis. Finally, the platelet abruptly terminates contractility and detaches from the surface."

The authors make a very important observation here, which in my opinion needs to be further elaborated during manuscript revision in the context of the link between the magnitude of force

generation during platelet activation by spreading and the dynamic mechanotransduction processes involving platelet actomyosin contractility/depolymerization of marginal band tubulin ring.

An interesting future follow-up topic of translational value could be an investigation using SIM-MFM with platelets having defective in actomyosin contractility or Wiskott-Aldrich syndrome (WAS) patient platelets.

5. Since the current experiments were performed under static conditions, can the authors comment on whether platelet adhesion under hydrodynamic shear may alter the force orientation vector of the integrin receptor based on the shear rate?

with best wishes,
Raghavendra Palankar

P.S. I must admit that I thoroughly enjoyed reading/reviewing this study and I believe the authors had fun performing these experiments themselves.

Reviewer #2:

Remarks to the Author:

In this paper, Blanchard et al. presents a novel method to map the cell receptor force orientation and magnitude with commercial SIM system. Taking advantage of the polarization super-resolution imaging capability of the commercial SIM system, the authors have upgraded their 3D orientation mapping technique to super-resolution force mapping. As biological force mapping is crucial to understanding a series of interactions, this work can be potentially deployed widely for studying the cell force at super-resolution. Overall, the authors present their idea clearly and the manuscript is well-written. However, the super-resolution results should be further optimized to eliminate factors that could be misleading. And, the advantages of using SIM as a super-resolution imaging technique, should be demonstrated. I have the following comments:

1. My biggest concern is that, despite the spatial super-resolution offered by SIM, the SIM-MFM does not show new insights or significantly better results over conventional MFM. Is this because of the poor SNR which obstacles the resolution/accuracy, or it reflects that the MFM field does not have super-resolved feature beyond diffraction limit? This should be clearly distinguished.
2. This paper is based on the recent polarization SIM work on Nature Communications (Ref. 44). Here in this work, the axial orientation of the 2D dipole (or the "disk" dipole) is inferred from the polarization modulation depth, which is also related to the extinction ratio of the polarized excitation. The authors need to measure the polarization states of the excitations as suggested in Ref. 44, and to discuss whether depolarization affects their measurement accuracies.
3. The uneven illumination is calibrated with the thick fluorescent slides, however, the illumination pattern will possibly change when changing the focus. Therefore, we suggest the authors to calibrate the uneven illumination with fluorescent beads or with a thin layer of fluorophore.
4. Although the spatial resolution of SIM-MFM is doubled to reach 100 nm, the polarization resolution of SIM-MFM is still diffraction limited. The authors should revise line 466 in the Discussion.
5. The raw data in Figure 2 seems to have low signal-to-noise ratio, and SIM reconstruction artifacts are obvious in Figure 2 and Figure 3. In the super-resolution image processing part of SI (line 191), the author claim $NA=1.4$ but the system NA is 1.49. The wavelength should be the emission wavelength not the excitation wavelength.
6. The raw data should be evaluated with SIMcheck and the reconstruction parameters should be further optimized to reduce reconstruction artifacts. In Fig. 3d, the SIM super-resolution result has shown apparent illumination pattern at one single direction. I suggest the authors to check the raw data with SIMcheck, to see the modulation depth, and distribution between the diffraction orders

are within a reasonable range. Normally this should not happen if the system is well calibrated.

Likewise, the Fig. 3b shows strong dotted noise, which might be partially due to low SNR. And Fig. 3e3 shows severe honeycomb structure across the entire image. I suggest the authors to increase the imaging time for a better SNR and better reconstruction.

7. Why the error in orientation measurement is larger in SIM-MFM than fluorescence polarization microscopy in Ref. 16? The author think it may cause by illumination intensity fluctuation, so further image pre-processing/calibration should be performed to improve measurement accuracy.

8. The calculated tilt angle θ seem to be around 45° and have little difference in different samples or different regions of the same sample. The effectiveness of the method of tilt angle calculation should be further evaluated.

9. In Fig 5 and Fig. 6, we could not intuitively see what new information the SIM has brought after the resolution enhanced. It seems that these analyses could be completed on the MFM only. The author should compare the analysis results of SIM-MFM and MFM. And, are Fig 5a displaying wide-field images? It does not look like a super-resolution image reconstructed by SIM.

10. The equation 6a contains 17, 77, and 137 degrees. These values are due to the angle between the illumination and the CCD detectors, making the expression only applies to this very specific case. To make it general, I suggest the authors to change them to 0, $\pi/3$, and $2\pi/3$.

11. Eq. (8) lacks a physical model, reference, or a well-defined explanation.

12. In Fig. 6b, why the three data sampling spots have different intervals?

13. In Fig. 6 caption, $\langle A \rangle / (\langle c \rangle + 1)$ is very hard to follow. From Fig. 6b, A is greater than 1 and c is less than 1. So it is unclear to me how it is normalized. And, the figure legend of Fig. 6c is expressed as $\langle A \rangle / \langle c + 1 \rangle$.

Minor comments:

1. Figure 2 and Figure 6 legend missing the description of scale bar.
2. Some letters in Figure 2 legend are capitalized, such as B,C,F,G.
3. The platelet results in Figure 6a looks overexposure and the contrast should be readjusted. Figure 6a also miss the intensity bar.
4. Line 335-336, cross-reference error appears.
5. Line 631, format error in reference 55 and 56.

Reviewer #3:

Remarks to the Author:

In this study the authors combine DNA-hairpin based fluorescent molecular force sensors with polarized structured illumination microscopy (SIM) to generate traction maps of platelets, 3T3 mouse fibroblasts, and mouse T-cells adhering to glass coverslips. Polarized SIM yields traction maps with approximately 100 nm spatial resolution, along with estimates of the polar and azimuthal angles of the force vector. The authors report that "platelets dynamically re-arrange the orientation of their integrin forces during activation," that the focal adhesions of fibroblasts have subregions with different levels of traction stress, and forces in T-cells adhering to an anti-CD3 functionalized surface are not polarized.

Summary: Unfortunately the paper is not up to the standards of Nature Communications, and more broadly is not ready of publication. The technical advance reported here is potentially useful but incremental. A few aspects of the data analysis require clarification and improvement. The presentation of the work is not consistent with scholarly standards.

1) While the combination of MTFM and polarized SIM is potentially helpful but incremental: other research groups have used superresolution techniques to improve the spatial resolution of TFM and related measurements (see below). For that reason I feel that this paper is better suited to a specialized journal, for example Biophysical Journal, Cytoskeleton, or perhaps the Journal of Microscopy.

2) The authors need to present a balanced analysis of prior work in the field. For example, the spatial resolution of traction force microscopy is, not surprisingly, improved when the technique is combined with superresolution microscopy techniques: see for example Stubb et al. *Nano Lett* 2020, Colin-York et al. *Nano Lett.* 2019, and Colin-York *Nano Lett* 2016. Protein-based fluorescent sensors have been combined with superresolution imaging to yield traction maps with comparable spatial resolution to those reported here (Morimatsu et al. *Nano Lett* 2015). Finally, it is worth noting that multiple prior publications demonstrate the presence of subregions within focal adhesions, Hu et al. *Cytoskeleton* (2015) being an early example.

3) Relatedly, the authors have a tendency to hype their work. The use of the term “turn-key” in the title is arguable: all of the studies cited above used microscopy technology that is readily available in commercial form (SIM, STED, or TIRF) and could therefore be termed “turn-key” by the same logic. The claim that MTFM yields a “three orders-of-magnitude improvement in force magnitude resolution” is not supported. The technique averages over all of the DNA hairpins in a pixel, just as TFM averages over the forces exerted by integrins within a given area. Further, MTFM is not comparable to TFM in important ways: because the signal from the DNA hairpins is switch-like, it does not provide accurate local measurements of local stresses. Further, there is a two-fold degeneracy in the measurement of the azimuthal (ϕ) angle, a limitation that is only indirectly acknowledged in the discussion. The authors do not provide a rigorous estimate of the spatial resolution of their force maps, but instead rely on the best-case resolution for SIM as a proxy. The authors’ tendency to over-sell their work does a disservice both to their field and, over the longer term, to themselves.

4) The authors should provide evidence that the SIM maps presented are not subject to reconstruction artifacts. For example, the reticular structures observed in Figure 3e are reminiscent of the artifacts previously noted when using Weiner deconvolution (Fan et al. *Biophys Rep.* 2019) and low signal intensities.

5) The authors report that Monte Carlo simulation suggests that estimates for the polar angle θ are poor at low photon counts—presumably this is why they discard low intensity pixels in their analysis. For this reason it is important that they quantify and report the distribution of photon counts and the cutoff applied for their analyses.

6) The polar angle for the force vector of 45 degrees measured for 3T3s seems large relative to previous measurements (Liu *PNAS* 2015). It seems plausible that this may reflect the tendency of the measurement used here to overestimate low θ values (Fig. S5E). Liu et al. is, incidentally, another relevant prior publication that is not cited.

7) The rationale for splitting platelets into “two groups (28 increasing alignment cells and 22 cells with non-increasing alignment)” is not clear. The authors should provide evidence that these cells are distinguishable in an independent, biologically relevant way. Otherwise, the parsimonious interpretation is that all the cells come from the same distribution, of which half happen to fall above the measurement threshold for force vector alignment. This ambiguity makes this section of the paper difficult to evaluate.

8) The null result reported for T-cells (no obvious force polarization) is likewise difficult to interpret. The three possibilities listed by the authors are reasonable. However, it also seems possible that the force threshold for the sensor used may be too low or too high to measure the forces that are most relevant for T-cell activation. Another possibility is that the choice of antibody might matter: tangential force transmitted through 17A2 resulted in T-cell activation, whereas other mABs did not (Kim et al. *JBC* 2009). (So far as I can tell the authors do not report what antibody they used.) All of these ambiguities make this section of the paper likewise difficult to interpret.

9) The method used to derive confidence bands in Figure 5h is ad hoc, and needs to be replaced with a more rigorous approach.

Reviewer 1:

Reviewer summary: The manuscript by Blanchard et al. is a systematically designed, meticulously performed, and well-written proof-of-concept study on the feasibility of 3D-SIM in combination with molecular force microscopy to assess forces generated at receptor-ligand interfaces and importantly nanoscale organization and orientation of these forces based on transition dipole moments of fluorophore-conjugated to a DNA tension probe. In my opinion, the entire study documented in this paper of great interest not only to the light microscopy imaging community but across broad interdisciplinary fields spanning cell biology, mechanobiology/mechanchemistry, and biophysics. Blanchard et al. using SIM-MFM, provide biologically relevant key takeaway messages with their studies on human blood platelets and primary T-cells from mouse and in-vitro cell line such as the 3T3 fibroblast, supported by solid datasets and detailed analysis. In addition, the authors also provide extensive details on the implementation of SIM-MFM for potential future users. In summary, the current version of the manuscript needs no additional experiments to substantiate the findings as observations and the interpretations thereof are well within the scope of the manuscript. While I am limiting myself to the platelet part of the manuscript, my specific queries to the authors are as follows:

Response: We thank the reviewer for their favorable and insightful review of our work. We also thank the reviewer for their numerous helpful suggestions for future work.

Comment 1: 1. Imaging protocols, parameters, and data analysis: I would highly recommend the authors to prepare a step-by-step protocol with a visual guide on the implementation of SIM-MFM on the Nikon N-SIM system for image acquisition and post-acquisition image processing steps and publish this into a citable repository such as the Nature Protocol Exchange. In my opinion, these protocols form the core of this paper and hence will be relevant for users who intend to start testing SIM-MFM on their own.

Response: We greatly appreciate this suggestion and have begun preparing a protocol for publication on the Nature Protocol Exchange. We believe that the methods section is sufficiently detailed for SIM users to implement SIM-MFM, but we agree that a detailed protocol that walks researchers through all steps – from probe synthesis to image processing – would greatly improve the ease-of-entry to this research area. Unfortunately, due to time and personnel constraints, we were not able to submit the protocol prior to resubmission of this work. Notably, our group has recently submitted a protocol to JOVE that describes the surface preparation protocol used in this work¹. We have included a citation to this article in the revised manuscript.

Comment 2: Can the authors comment on the influence of changes in the local microenvironment at the ligand-receptor interface on the plasma membrane especially in platelets that actively secrete their acidic granular content during spreading on the transition dipole moment for Cy3b and how it may influence ϕ , θ and I_{max} ? Are there other fluorophores that show similar transition dipole moment as Cy3b? If yes, is it possible to combine two or more DNA tension probes with unique ligands to target different receptors within the same experiment? or are there limitations? Please discuss.

Response: We thank the reviewer for these thoughtful considerations. Based on previous studies, cyanine dyes – which have a wide range of excitation and emission wavelengths – in general are thought to exhibit the base-stacking interactions necessary for SIM-MFM^{2,3}. In contrast, we previously tested Alexa dyes and showed that these dyes do not align their transition dipole moments with the nucleobases⁴. Several studies have also shown that, the fluorescence intensity of cyanine dyes such as Cy3b are still maintained at acidic conditions with pH 4-6 (e.g. see https://help.lumiprobe.com/p/44/fluorescence_cyanine_dyes), similar to the pH of stress granules. Therefore, we find it unlikely that degranulation would alter fluorescence anisotropy. Moreover, degranulation events are highly transient, and pH should rapidly reach equilibrium with surrounding media over subsecond times scales⁵.

We have expanded upon our discussion section to acknowledge that there are many unknowns regarding the local microenvironment under the cell (see response to comment 3, below). We have also included new commentary on the potential of imaging multiple probes with distinct fluorophores simultaneously:

“...of molecular force orientation. In the future, it may be possible to perform two-color SIM-MFM using probes with other duplex terminus-stacking dyes such as cyanine 5 (Cy5). Such an advance could enable simultaneous mapping of forces applied through probes that bind to different surfaces receptors, or probes that exhibited different $F_{1/2}$ values. We expect that ongoing...”

Comment 3: In the current study authors used cRGDfK as a high-affinity ligand for $\alpha v\beta 3$ and $\alpha 5\beta 1$ integrin which also bind to fibronectin with high affinity. Previously it has been shown that platelets secrete or “self-deposit” their own ECM proteins such as fibronectin (and maybe other ligands) while spreading in Vitro and this potentiates $\alpha I I b\beta 3$ mediated platelet adhesion/spreading beyond the available ligand covered surface. (PMID: 25964667). Can the authors provide additional comments on how this phenomenon may influence the current observations on the observed differences in force tension points and their orientation along lamellipodia and in the inner regions of a spreading platelet during the course of timelapse measurements?

Response: We appreciate the reviewer for bringing these fascinating observations to our attention and have briefly commented upon them in our discussion:

“This finding raises important questions about the dynamic processes governing platelet mechanics. Future work could use timelapse SIM-MFM, along with two-color imaging of structural proteins such as tubulin, actin, and vinculin⁶, to investigate relationships between cytoskeletal network re-arrangement, focal adhesion growth, and tension alignment. Moreover, SIM-MFM could be a powerful tool for studying platelet-related diseases such as Glanzmann thrombasthenia or Wiskott-Aldrich syndrome (in which actomyosin contractility is impeded)⁷. It may also be necessary to investigate the dynamic effects of platelet biochemical activity on tension probe-functionalized surfaces; it was previously shown that platelets secrete ECM proteins such as fibronectin while spreading in vitro⁸, which can potentiate $\alpha I I b\beta 3$ mediated adhesion. Because cRGDfK is specific to $\alpha v\beta 3$ and $\alpha 5\beta 1$, the gradual potentiation of $\alpha I I b\beta 3$ signaling (driven by the accumulation of ECM proteins secreted onto the surface surrounding/below platelets) could introduce temporal effects on platelet mechanics. Finally, SIM-MFM is well-suited to investigating the effects of physiologically-relevant shear flow rates on platelet mechanics and dynamics⁹.”

In addition, we would like to note that, in previous work, we added exogenous fluorescent fibronectin to platelets and found no diminishment of tension signal¹⁰. These findings suggest that secreted ECM minimally alters our observations of $\alpha v\beta 3$ and $\alpha 5\beta 1$ tension, potentially because integrins are sufficiently abundant to simultaneously bind to secreted ECM molecules and tension probes. That said, we passivate our surface for 45 minutes with 1 mg/mL BSA (see methods) to minimize the adsorption of exogenous proteins and ECM.

Comment 4: To quote the authors "An interesting finding of this analysis is that spreading occurs on a significantly faster timescale than alignment; $\tau_{\text{spread}}=1.7$ min, while $\tau_{\text{align}}=5.0$ min ($p<0.001$, Wilcoxon rank-sum test) (Fig. 5g). This result suggests that platelet activation displays a characteristic mechanical progression that includes three phases. In the first phase, the platelet spreading area and the traction forces grow until reaching a steady-state plateau. In the second phase, forces within the platelet

re-organize and realign along an axis. Finally, the platelet abruptly terminates contractility and detaches from the surface." The authors make a very important observation here, which in my opinion needs to be further elaborated during manuscript revision in the context of the link between the magnitude of force generation during platelet activation by spreading and the dynamic mechanotransduction processes involving platelet actomyosin contractility/depolymerization of marginal band tubulin ring. An interesting future follow-up topic of translational value could be an investigation using SIM-MFM with platelets having defective in actomyosin contractility or Wiskott-Aldrich syndrome (WAS) patient platelets.

Response: We thank the reviewer for this comment and for recognizing the significance our observation. In addition to an expanded discussion addressing some of these points (see response to comment 3, above), we have expanded the relevant results section to include mention of these processes, which we hope to investigate in greater detail in future work:

"...study of platelet mechanics. In future work, two-color SIM-MFM may prove valuable for revealing the relationship between tension alignment and the reorganization of the tubulin and actin cytoskeletal networks. What are the roles of tubulin marginal band coiling and actomyosin contractility in controlling the dynamics of platelet mechanics? How do membrane structures such as focal adhesions change over time and modulate traction? These questions could be answered in part using SIM-MFM in parallel with z-stack imaging of cell-permeable stains incubated with live platelets (e.g. Tubulin Tracker, made by Molecular Probes, Eugene, OR, USA).

Notably, a previous TFM-based ..."

As a preview for upcoming work, we have included a preliminary set of images below. An RICM image shows locations of close contact between two platelets and a tension-probe functionalized surface. A fluorescent, cell-permeable tubulin stain (Tubulin Tracker) shows that the tubulin networks in both platelets have largely collapsed at the time of acquisition, colocalizing primarily to the tension-free centroid of the platelets. It also appears that the collapsed tubulin network is ellipsoidal in shape, with the long axis of the ellipsoid parallel to what appears to be the axis of tension. The dynamics of tubulin re-arrangement in relation to the dynamics of spreading and the dynamics of tension alignment will be the subject of future work. Furthermore, a similar stain for actin will allow near-simultaneous visualization of tension and both cytoskeletal networks.

Comment 5: Since the current experiments were performed under static conditions, can the authors comment on whether platelet adhesion under hydrodynamic shear may alter the force orientation vector of the integrin receptor based on the shear rate?

Response: We agree with the reviewer that the effect of flow would be interesting to investigate using SIM-MFM. We have added commentary and cited relevant literature to the discussion section at the end of the manuscript (see response to comment 3, above).

Reviewer 2:

Reviewer summary: In this paper, Blanchard et al. presents a novel method to map the cell receptor force orientation and magnitude with commercial SIM system. Taking advantage of the polarization super-resolution imaging capability of the commercial SIM system, the authors have upgraded their 3D orientation mapping technique to super-resolution force mapping. As biological force mapping is crucial to understanding a series of interactions, this work can be potentially deployed widely for studying the cell force at super-resolution. Overall, the authors present their idea clearly and the manuscript is well-written. However, the super-resolution results should be further optimized to eliminate factors that could be misleading. And, the advantages of using SIM as a super-resolution imaging technique, should be demonstrated. I have the following comments:

Response: We thank the reviewer for their favorable assessment of our work, and for recognizing the potential for our methods to be widely-deployed. Based on the reviewer's suggestions, we have implemented several updates detailed below.

Comment 1: My biggest concern is that, despite the spatial super-resolution offered by SIM, the SIM-MFM does not show new insights or significantly better results over conventional MFM. Is this because of the poor SNR which obstacles the resolution/accuracy, or it reflects that the MFM field does not have super-resolved feature beyond diffraction limit? This should be clearly distinguished.

Response: To better illustrate the improved spatial resolution of SIM-MFM, we have conducted a more detailed analysis summarized in new supplemental figures. The new figure illustrates a systematic, automated method for comparing the resolution of conventional MFM and SIM-MFM using the lamellipodial edge of platelet tension as the common structure being compared. The figure also shows that applying this method to a set of 19 platelets consistently shows a significant enhancement in spatial resolution by ~82 nm, which is similar to (but less than) the expected ideal enhancement of ~95 nm:

a) Depiction of procedure used to obtain intensity linescans around the lamellipodial edge of the platelet. First, a polygon was traced around the platelet's perimeter using MATLAB's built-in `getpts` function. Then 500 points were interpolated around the perimeter of the closed polygon and the points were smoothed with a ten-point filter. At each of the 500 points, an intensity linescan perpendicular to the polygon's perimeter extending ~ 378 nm (6 WF pixels) in each direction was obtained using MATLAB's built-in `improfile` function. **b)** A Gaussian function was

then fit to each linescan using MATLAB's built-in "fit" function. Four representative linescans are shown, with both WF (blue) and SIM (red) shown. **c**) The full-width half maximum (FWHM) was then obtained from each Gaussian. **d**) Two histograms show the distribution of FWHM values recorded by repeating this process for each of the 500 linescans. Note that linescans were discarded if they did not meet all of the following conditions: either best-fit sinusoid's maximum is located at least 315 nm away from the linescan's midpoint; or either of the best-fits sinusoid has a maximum height of less than 0.2 (intensity units in these images are background-normalized). **e**) Nineteen pairs of SIM-MFM orientation colormaps are shown (for each pair, WF is on the left and SIM is on the right). Dotted white lines in each WF image show user-drawn and smoothed lamellipodial edge polygons used for drawing linescans. White lines in top left corners of images denote scale bar of 2 μm . **f**) A boxplot showing the change in the average deviation angle (**Fig. S4**) of pixels within ~ 378 nm of the lamellipodial edge polygon when calculations are weighted using widefield MFM, rather than super-resolution reconstruction images. Each grey circle denotes the average deviation angle change from one of the platelets in the analysis. We observe a small, but significant improvement of ~ 1.5 degrees when super-resolution reconstructions are used. Red shading denotes standard error of the mean and purple denotes 95% confidence interval. **g**) Violin plots showing the distribution of FWHM values from linescans of SIM (yellow) and WF (blue) images. These plots show that the FWHM is consistently lower for SIM reconstructions compared to WF for all 19 platelets included in this analysis. **h**) Box plots showing the median FWHM values for all 19 platelets, showing an improvement of ~ 80 nm from ~ 295 nm to ~ 213 nm when super-resolution reconstructions are used. Red shading denotes standard error of the mean and purple denotes 95% confidence interval. Under optimal conditions, the WF resolution of 561 nm light with $NA=1.49$ (as is the case with our microscope) is:

$$FWHM_{WF} = \frac{(0.51)(561 \text{ nm})}{1.49} = 191 \text{ nm}$$

A two-fold improvement, as is generally expected for ideal implementations of SIM, would thus produce $FWHM_{SIM} = 96$ nm, which is a resolution improvement of ~ 95 nm. As such, our implementation of SIM reconstruction with SIM-MFM, which yields an ~ 82 nm improvement in resolution, comes close to the ideal resolution enhancement.

In order to highlight the quantitative improvement afforded by SIM-MFM, we have also added the boxplot shown in subfigure **h** to Fig. 2. In addition, to offer more examples of the spatial resolution improvement offered by SIM-MFM, we have included a new supplemental figure (**Fig. S7**) that shows additional images of 3T3 fibroblasts in widefield and super-resolution. These additional examples also include linescans that demonstrate the ability to resolve of additional features. These images provide qualitative depictions of resolution enhancements, complementing the quantitative proof in **Fig. S6**.

However, as the reviewer mentions the SNR for tension maps is indeed weaker than that obtained for past work using fixed and stained cytoskeletal structures. Our data are all collected on live dynamic cells, which offers exciting opportunities in studying dynamics but inevitably results in lower SNR, particularly because of the highly transient nature of biophysical forces in cells. In our response to comment 6 below, we discuss our use of SIMcheck to evaluate our data, which revealed that our SIM-MFM acquisitions are in the "low-to-moderate quality" range. Accordingly, improvement of the signal to noise is a crucial target area of future MFM development. In the discussion section of our revised manuscript, we now discuss these limitations and future directions:

The moderate signal to noise ratio (SNR) is a key factor limiting super-resolution reconstruction of SIM-MFM images. The primary factor constraining SNR is photobleaching of the tension probes' fluorophores; the need for several high-quality images per SIM acquisition (and the need for several acquisitions during timelapse imaging), means that moderate laser powers and short exposure times must be used to ensure that substantial photobleaching does not occur between images. Accordingly, techniques that improve photostability (e.g. the incorporation of DNA fluorocubes¹¹ and/or the use of in-solution oxygen scavenging systems¹²) could enable higher SNR, longer imaging durations, and higher frame rates

Finally, we would also like to emphasize that the attainment of super-resolution is not the sole advance reported in this manuscript; there are many features of SIM-MFM highlighted in our manuscript, including the ability to use SIM microscopes in turn-key fashion to implement MFM, the ability to pair SIM-MFM with SIM imaging of tagged biomolecules (e.g. GFP-paxillin, Fig. 3), and the unprecedented ability to acquire parallel MFM timelapses (**Fig. 5**). We also present novel biological findings about platelet dynamics (**Fig. 5**) and the organization of T-cell forces (**Fig. 6**), which don't require super-resolution reconstruction. To better ensure that the breadth of these advances are accounted for, we have made edits to slightly de-emphasize the super-resolution aspect of our work, most notably by removing the term "super-resolution" from the title.

Comment 2: This paper is based on the recent polarization SIM work on Nature Communications (Ref. 44). Here in this work, the axial orientation of the 2D dipole (or the "disk" dipole) is inferred from the polarization modulation depth, which is also related to the extinction ratio of the polarized excitation. The authors need to measure the polarization states of the excitations as suggested In Ref. 44, and to discuss whether depolarization affects their measurement accuracies.

Response: We thank the reviewer for raising this concern. To address this concern, we have included a measurement (and a description of the measurement) of the extinction ratio in the methods section at the end of the subsection on structured illumination microscopy:

*"To confirm linear polarization of the excitation laser, a linear optical polarization filter (WP25M-VIS, Thorlabs, NJ) on a rotatable mount (RSP1/M, Thorlabs, NJ) was placed between the objective and an optical power meter (7Z02621, Ophir Starlite, Ophir Photonics, PN) and set to 532 nm detection mode. The polarization axis of the polarization filter was aligned normal to the polarization axis of light exiting the objective at 0° by rotating the mount until the reading by the power meter was minimized. The 532 nm laser was set to operate at 100% of its full power in SIM mode while the polarization filter was rotated in 20° increments, measuring the power of the light exiting the objective at each 20° increment. Three replicates of this experiment showed that the laser intensity varied sinusoidally with respect to polarizer angle (**Fig. S1**). The intensity minimizing at zero, thus confirming linear polarization of the laser."*

And included a new supplemental figure that illustrates full extinction when the polarizer and the excitation polarization are aligned, thus confirming that the excitation beam is linearly polarized:

Polarization verification was performed as described in the section paragraph of the methods subsection “**Structured illumination microscopy (SIM)**”. **a)** Microscope excitation path diagram adapted from Fig. 1a depicting arrangement of the optical power meter and rotatable linear polarizer. The polarization modulator (PM) and diffraction grating (DG) used to create the SIM striped interference pattern were removed from the light path for this measurement. **b)** Plots of three experimental replicate measurements at various linear polarizer angles. Black curve shows best-fit \sin^2 curve fit to the triplicate averages. Fitting was performed using MATLAB’s built-in `fmincon` function, and was constrained to a negative sinusoid minimum below zero. Accordingly, the best-fit sinusoid had a minimum of zero, corresponding to linearly-polarized light. Elliptically polarized light would have produced a positive sinusoid minimum.

We also include brief reference to this measurement in the main text:

“..previous implementation of MFM⁴ which required ~3.6 sec per acquisition. We also verified that the microscope’s illumination laser was linearly polarized using a photometer and a linear polarizer (Fig. S1). We next tested whether SIM could...”

In addition, our measurement shows that the excitation beam is nearly perfectly linearly polarized before interference, which is consistent with calibration performed by Zhanghao *et al.* in the article mentioned by the reviewer. The polarization at the sample plane cannot be measured as easily. Accordingly, Zhanghao *et al.* semi-quantitatively characterized the polarization depth in SIM mode by imaging a standard sample (phalloidin-labeled actin filaments in U2OS cells). They observed that the polarization factor was similar to what had been observed in previous studies of similar samples with verifiably linearly polarized excitation light. Their analysis suggested that switching the excitation beam to the interfering SIM mode did not significantly distort the polarization of their laser beam. We performed a similar control experiment by imaging DiI-labeled microspheres. In our previous MFM work, we also imaged this type of sample. We recorded similar polarization factors $((A + c)/c) \approx 1.5$ in both studies.

Comment 3: The uneven illumination is calibrated with the thick fluorescent slides, however, the

illumination pattern will possibly change when changing the focus. Therefore, we suggest the authors to calibrate the uneven illumination with fluorescent beads or with a thin layer of fluorophore.

Response: We thank the reviewer for this astute observation. As mentioned in our methods, we used two different techniques in this work:

“...images were generated on the day of the experiment by collecting at least six SIM acquisitions with the microscope focused onto the bottom of a Chroma slide or a surface with 100% open tension probes, which were prepared by adding a “opening strand” (which is complementary to the full hairpin) during the DNA strand hybridization step. Each acquisition ...”

While the first method (Chroma slide) may have some penetration depth-related issues as the reviewer points out, the second method is essentially a thin layer of fluorophore as suggested. However, the first method is much less labor intensive and produces much smoother images with reduced localized noise. To determine whether the two methods could be expected to produce similar results, we generated background illumination profiles using both methods from images collected on the same day. We then took the ratio of these two background illumination profiles. The resulting ratio was essentially a flat field with no noticeable global variations. This analysis suggests that both methods are equally effective at accounting for long-range variations in intensity (i.e. the Gaussian illumination profile), as well as variations between individual images. We have included a supplemental figure (**Fig. S5**) to demonstrate this analysis. We have also included a reference to this figure in the methods section.

Comment 4: Although the spatial resolution of SIM-MFM is doubled to reach 100 nm, the polarization resolution of SIM-MFM is still diffraction limited. The authors should revise line 466 in the Discussion.

Response: We thank the reviewer for raising this concern. This limitation is fundamental to pSIM-based techniques and was noted in the original pSIM paper, which was also published in *Nature Communications*¹³. While we did mention this point earlier in the text, we agree that it is important to

continue to emphasize this point because it may be easy for readers to miss. Therefore, we have updated our discussion to explicitly note this and other fundamental disadvantages of our technique:

“... A broader limitation of pSIM-based techniques such as SIM-MFM is that super-resolution reconstruction only improves the resolution of the intensity map – not the force orientation map. This limitation arises because the orientation of the polarization and the illumination striping are rotated simultaneously, rather than separately. In other words, the spatio-angular cross-harmonics cannot be detected with conventional SIM microscopes¹³. While it may be possible to engineer optical systems that can resolve cross-harmonics by modulating the polarization and diffraction patterns separately, it is unlikely that such acquisitions could be implemented using existing SIM microscope hardware. Finally, a key limitation of MFM is that force orientation measurement is degenerate ...”

Furthermore, to avoid misleading readers we have revised the opening of the discussion from:

“We have demonstrated that a SIM microscope can be used to implement MFM for rapid mapping of molecular force orientation at ~100 nm resolution. To demonstrate...”

to:

“We have demonstrated that a SIM microscope can be used to implement MFM for rapid mapping of molecular force orientation. The inherent super-resolution capability of SIM allows mapping of force magnitude information with ~100 nm resolution. To demonstrate...”

This modification specifies that the resolution enhancement is limited to information on force magnitude. In addition, this modification separates the two claims (turn-key MFM and super-resolution) from each other.

Comment 5: The raw data in Figure 2 seems to have low signal-to-noise ratio, and SIM reconstruction artifacts are obvious in Figure 2 and Figure 3. In the super-resolution image processing part of SI (line 191), the author claim NA=1.4 but the system NA is 1.49. The wavelength should be the emission wavelength not the excitation wavelength.

Response: During revisions, we switched our image reconstruction from FairSIM to Nikon Elements. The Nikon Elements software affords a greater degree of control and usability, which allowed us to obtain higher-quality reconstructions with fewer reconstruction artifacts. Accordingly, we have re-written this section and the parameters mentioned by the reviewer (NA and wavelength) are no longer listed (these parameters are saved directly into the image metadata and loaded into the Elements software automatically).

Comment 6: The raw data should be evaluated with SIMcheck and the reconstruction parameters should be further optimized to reduce reconstruction artifacts. In Fig. 3d, the SIM super-resolution result has shown apparent illumination pattern at one single direction. I suggest the authors to check the raw data with SIMcheck, to see the modulation depth, and distribution between the diffraction orders are within a reasonable range. Normally this should not happen if the system is well calibrated. Likewise, the Fig. 3b shows strong dotted noise, which might be partially due to low SNR. And Fig. 3e3 shows severe honeycomb structure across the entire image. I suggest the authors to increase the imaging time for a better SNR and better reconstruction.

Response: We were able to connect many of these errors back to our choice of image reconstruction software. Accordingly, we have switched from the ImageJ plugin FairSIM to the Nikon Elements

software, which offers greater control over reconstruction. Our updated reconstructions are much smoother, as evidenced by side-by-side views of Fig. 2f, which has fewer apparent honeycomb errors:

In addition, we have checked our images with SIMcheck. Example results from individual checks, as well as summary statistics collected from several cells, are now shown in Fig. S8:

While the plugin found that they exhibit “low-to-moderate” quality, we wish to re-emphasize that super-resolution imaging is not the sole (or primary) advance presented in this paper. Also, note that our images are collected using live, mechanically active cells, whereas SIM imaging is often performed on fixed cells.

The limited quality is due in large part to low SNR (which is a byproduct of working with live dynamic cells). SNR can be improved with increased imaging time as suggested by the reviewer. However, doing so increases the rate of photobleaching, which also reduces the quality of reconstruction. Furthermore, because background noise in MFM arises due to the relatively bright background signal from unopened tension probes (rather than camera read noise), our background signal scales with exposure time. Our image acquisition routine was optimized to balance SNR with photobleaching, and so the results

presented here are close to optimal. We discuss these limitations and potential solutions in the discussion section of our updated dataset (see response to comment 1).

Comment 7: Why the error in orientation measurement is larger in SIM-MFM than fluorescence polarization microscopy in Ref. 16? The author think it may cause by illumination intensity fluctuation, so further image pre-processing/calibration should be performed to improve measurement accuracy.

Response: We thank the reviewer for raising this concern. We believe the primary reason for increased error is the reduced number of distinct α values used in this work (3) compared to previous work (72). To better convey this point, we added the following:

“...resulted in at most a 7% deviation from the ideal. These errors likely result from the small number of α angles (3) relative to our previous implementation of MFM (72), and in future work it may be possible to address this issue by implementing SIM-MFM with more than 3 α angles. Nonetheless, we do not expect this form of systematic error, which is small in magnitude, to meaningfully bias our measurements.

We would also like to note that, while the errors in this work are larger than in previous work, they are still small in magnitude. For example, the histogram in Fig. 4b is truncated, but if the full scale of the plot is shown, it is clear that ϕ angles are for the most part uniformly distributed, with only small biases:

These small errors are not substantial enough to alter the important conclusions that we draw in this work (highlighted in Figs. 5-6), which rely on the type of ensemble level estimates that would be insensitive to the systematic errors shown here.

Comment 8: The calculated tilt angle θ seem to be around 45° and have little difference in different samples or different regions of the same sample. The effectiveness of the method of tilt angle calculation should be further evaluated.

Response: We thank the reviewer for raising this concern. In fact, we have similar concerns about the effectiveness of the θ measurement; in previous work⁴, we performed modeling that suggested that θ can be systematically biased by 1) local heterogeneity, and/or 2) thermal fluctuations of the probe at low force magnitude. These effects, which both can cause θ to be systematically underestimated due to increased orientational heterogeneity, are separate from the photon noise-based effects that we investigated in this work. However, we neglected to address these points in our original submission. These effects are not expected to systematically bias ϕ calculations⁴. For this reason, the majority of our analyses centered on our measurements of ϕ . In future work, we do have intentions to decouple orientational heterogeneity and θ using a technique that we developed called variable incidence angle linear dichroism (VALiD)¹⁴. However, as VALiD is not SIM-based, these future studies are outside the scope of this work.

In addition, please note that we previously found that θ varied slightly from the periphery to the centroid of platelets⁴. In a later study, we found that fibroblast integrins exhibited near-vertical forces on supported lipid bilayers (rather than glass)¹⁵. In addition, we found that T-cell receptor forces do not have a $\theta = 45^\circ$ value. Together these results show that the technique doesn't inherently produce $\theta = 45^\circ$ in all scenarios, and rather it may be a reflection of the biology as the forces measured in platelets and fibroblasts are mediated by integrins. Therefore, our findings (that θ is similar between fibroblasts and platelets) suggests that the geometry of integrin mechanics is similar across cell types⁴.

Comment 9: In Fig 5 and Fig. 6, we could not intuitively see what new information the SIM has brought after the resolution enhanced. It seems that these analyses could be completed on the MFM only. The author should compare the analysis results of SIM-MFM and MFM. And, are Fig 5a displaying wide-field images? It does not look like a super-resolution image reconstructed by SIM.

Response: We apologize for the confusion, Figs. 5 and 6 are not based on super-resolution reconstructions. Our work as a whole is intended to showcase the multiple capabilities of SIM-MFM, which includes super-resolution as well as timelapse imaging, two-color imaging, and the turn-key nature of SIM-MFM measurements. However, in response to the reviewer's requests we have included more detailed comparisons between widefield and super-resolution depictions of MFM (see response to comment 1).

Comment 10: The equation 6a contains 17, 77, and 137 degrees. These values are due to the angle between the illumination and the CCD detectors, making the expression only applies to this very specific case. To make it general, I suggest the authors to change them to 0, $\pi/3$, and $2\pi/3$.

Response: We thank the reviewer for this discussion. The derivation in supplemental note 1 includes generalized equations as requested. We have edited our manuscript (following equations 6 & 7) to better highlight the general equations:

"...applied these calculations to the entire image set. Note that Supplemental Note 1 includes calculations that are generalizable to SIM systems with initial α values other than 77° . Our initial results reproduced..."

Comment 11: Eq. (8) lacks a physical model, reference, or a well-defined explanation.

Response: We thank the reviewer for bringing this deficiency to our attention. We have added an explanation of this equation in the paragraph following the equation:

*"...the unit step function that denotes initiation of spreading at $t = t_{attach}$ (**Fig. 5b,c**). We constructed this equation – which is based purely off of our observation of the nature of the data – such that the first term (one minus the decaying exponential) reflects an asymptotic increase in T , while the second term (with the exponential in the denominator) is a sigmoid function that reflects an s-shaped decrease in T to baseline. We set T_{max} , t_{attach} , t_{detach} , $\tau_{retract}$, and τ_{spread} all as fit parameters..."*

Comment 12: In Fig. 6b, why the three data sampling spots have different intervals?

Response: We apologize for the confusion. Figure 6c is simply a re-sized version of Figure 1c. Accordingly, the three points represent 3 intensity vs. α values from a pixel indicated in Fig. 2c. We included a copy of this plot simply to illustrate what A and c mean, to aid with interpretations of the rest of the figure.

Comment 13: In Fig. 6 caption, $/(c+1)$ is very hard to follow. From Fig. 6b, A is greater than 1 and c is less than 1. So it is unclear to me how it is normalized. And, the figure legend of Fig. 6c is expressed as

/c+1>.

Response: We apologize for the confusion. We have an expanded description of this analysis in supplemental note 2, which is referenced in the main text. To better highlight this discussion, we have also added a reference to supplemental note 2 in the figure caption as well:

“...background close to the cell. A more detailed description of this analysis can be found in Supplemental Note 2. While platelets exhibit...”

In addition, we have changed /(<c>+1) in the caption to /c+1>, for clarity (the two are equivalent mathematically).

- Minor Comments 1-5:**
1. Figure 2 and Figure 6 legend missing the description of scale bar.
 2. Some letters in Figure 2 legend are capitalized, such as B,C,F,G.
 3. The platelet results in Figure 6a looks overexposure and the contrast should be readjusted. Figure 6a also miss the intensity bar.
 4. Line 335-336, cross-reference error appears.
 5. Line 631, format error in reference 55 and 56.

Response: We thank the reviewer for pointing out these oversights. We have fixed all of these issues in the revised manuscript, except for the overexposure comment in point 3. The intensity limits were chosen to be the same both for platelets and T-cells while enabling clear visualization of both. Because platelet signal is slightly brighter than T-cell signal, this results in the image appearing to be saturated for the platelet images.

Reviewer 3:

Reviewer summary: In this study the authors combine DNA-hairpin based fluorescent molecular force sensors with polarized structured illumination microscopy (SIM) to generate traction maps of platelets, 3T3 mouse fibroblasts, and mouse T-cells adhering to glass coverslips. Polarized SIM yields traction maps with approximately 100 nm spatial resolution, along with estimates of the polar and azimuthal angles of the force vector. The authors report that “platelets dynamically re-arrange the orientation of their integrin forces during activation,” that the focal adhesions of fibroblasts have subregions with different levels of traction stress, and forces in T-cells adhering to an anti-CD3 functionalized surface are not polarized. Unfortunately the paper is not up to the standards of Nature Communications, and more broadly is not ready of publication. The technical advance reported here is potentially useful but incremental. A few aspects of the data analysis require clarification and improvement. The presentation of the work is not consistent with scholarly standards.

Comment 1: While the combination of MTFM and polarized SIM is potentially helpful but incremental: other research groups have used superresolution techniques to improve the spatial resolution of TFM and related measurements (see below). For that reason I feel that this paper is better suited to a specialized journal, for example Biophysical Journal, Cytoskeleton, or perhaps the Journal of Microscopy.

The authors need to present a balanced analysis of prior work in the field. For example, the spatial resolution of traction force microscopy is, not surprisingly, improved when the technique is combined with superresolution microscopy techniques: see for example Stubb et al. Nano Lett 2020, Colin-York et al. Nano Lett. 2019, and Colin-York Nano Lett 2016. Protein-based fluorescent sensors have been combined with superresolution imaging to yield traction maps with comparable spatial resolution to those reported here (Morimatsu et al. Nano Lett 2015). Finally, it is worth noting that multiple prior publications demonstrate the presence of subregions within focal adhesions, Hu et al. Cytoskeleton (2015) being an early example.

Response: We thank the reviewer for raising this concern and apologize for presenting what came across as an unbalanced view of the literature. We have presented an expanded description of existing literature in the revised manuscript (see below). We have also added the reviewer’s reference to Hu et al. Cytoskeleton (2015) and a description of protein-based tension sensors – which we excluded because they have not been used to measure force orientation – in our introduction:

“As stated above, enhancing the spatial resolution of force mapping is desirable due to the nanoscale size of many mechanically active structures of interest. While we recently developed a method for localization of integrin forces with a spatial resolution of ~20 nm, this method does not capture force orientation information¹⁸. Similarly, while protein-based probes have recently been used for super-resolution force mapping, such probes have not yet been shown to be capable of reporting force orientation information⁴⁴⁻⁴⁸. Super resolution imaging has been employed to improve the spatial resolution of TFM⁴⁹⁻⁵¹, but even under ideal theoretical circumstances the spatial resolution is not expected to exceed ~500 nm due to TFM’s inherent need for substrate deformation⁵⁰. Accordingly, improving the spatial resolution of MFM would enable measurements of force orientation with unprecedented resolution (see Table S2 for a comparison of high resolution TFM and MTFM techniques). “

We have also noted previous literature showing nanoscale organization within focal adhesions, including a reference to the paper mentioned by the reviewer, to our results section:

“... The observation of these sub-regions within FAs, which has been borne out in previous literature¹⁶⁻¹⁸, can also be seen in additional examples in Fig. S7.”

We nonetheless stand by our original description of the limitations of traction force microscopy (TFM). It is true that advances have been made to increase the spatial resolution of TFM. However, TFM's spatial resolution, even in the best *theoretical* circumstances, is 2-fold worse than the diffraction limited resolution that we push past in this work. For example, Colin-York et al Nano Letters 2016 suggests a theoretical maximum resolution of ~500 nm, while the diffraction limit in our context is ~250 nm. In practice, the discrepancy is even greater because these theoretical limits have not been reached in experimental contexts. The fundamental basis for this limitation lies in the fact the TFM has two factors limiting spatial resolution: 1) diffraction of light, and 2) the propagation of mechanical strain through the soft underlying substrate. While super-resolution approaches can address the diffraction issue, they cannot address the softness issue, which is a fundamental requirement for TFM. In contrast MFM's spatial resolution is only limited by diffraction, so super-resolution techniques can push the spatial resolution down beyond the diffraction limit and even to tens of nanometers. Indeed, we are currently combining our recently-presented DNA PAINT-based approach with SIM-MFM to enable force orientation mapping of single molecule forces with spatial resolution on the order of 40 nanometers:

In addition to our brief description of these factors, we have compiled a systematic analysis of TFM literature and included it as supplemental table (Table S2) in the updated manuscript. This analysis, which builds on analyses presented in other recent works^{19,20}, was focused on papers that use super-resolution microscopy to improve the spatial resolution of TFM, including those referenced by the reviewer. Although these papers generally do not explicitly quantify their spatial resolution, we carefully obtained best possible estimates from each paper and found that the highest spatial resolution presented to-date was roughly 1 μm .

We do not wish to suggest that MFM will replace TFM, as the techniques are complementary. TFM reports global traction forces and provides the crucial ability to study cells on substrates with physiologically-relevant stiffnesses. However, when it comes to resolving forces generated by nanoscale structures, MFM can perform tasks that TFM fundamentally cannot.

	Study	Tension map spatial resolution*	Orientation resolvable?	Acquisition Time*	Required microscope	Note
TFM techniques	Sabass et al., 2008, Biophysical Journal ⁸	1 μm	Yes	1 s	Confocal	Two color beads
	Colin-York et al., 2016, Nano Letters ⁹	1-1.2 μm	Yes	20 s, 5 s	STED, Confocal	Modeling suggests 500 nm resolution is possible
	Colin-York et al., 2019, Nano Letters ¹⁰	1 μm	Yes	10 s	SIM	Acquisition time increased by need to collect z-stack
	Stubb et al., 2020, Nano Letters ¹¹	1-2 μm	Yes	20 s	Spinning-disk confocal or widefield microscope	
MTFM and MFM techniques	Zhang et al., 2014 Nature Communications ⁷	250 nm	No	0.1-1 s	Standard fluorescence microscope	Diffraction limited
	Brockman & Su et al., 2020 Nature Methods ¹	20 nm	No	1-10 min	Standard fluorescence microscope	DNA-PAINT based technique
	Brockman & Blanchard et al., Nature Methods ¹²	200 nm	Yes**	3.6 s	Fluorescence polarization microscope	Original MFM paper
	This work , SIM-MFM	110 nm	Yes**	1 s	SIM	Orientation map has diffraction limited resolution

* Spatial resolution and acquisition time values shown represent our best estimates. Not all studies reported these values explicitly.

** In existing MFM techniques, orientation measurements have a two-fold degeneracy that prevents unique force vector mapping

Comment 2: Relatedly, the authors have a tendency to hype their work. The use of the term “turn-key” in the title is arguable: all of the studies cited above used microscopy technology that is readily available in commercial form (SIM, STED, or TIRF) and could therefore be termed “turn-key” by the same logic. The claim that MTFM yields a “three orders-of-magnitude improvement in force magnitude resolution” is not supported. The technique averages over all of the DNA hairpins in a pixel, just as TFM averages over the forces exerted by integrins within a given area. Further, MTFM is not comparable to TFM in important ways: because the signal from the DNA hairpins is switch-like, it does not provide accurate local measurements of local stresses. Further, there is a two-fold degeneracy in the measurement of the azimuthal (ϕ) angle, a limitation that is only indirectly acknowledged in the discussion. The authors do not provide a rigorous estimate of the spatial resolution of their force maps, but instead rely on the best-case resolution for SIM as a proxy. The authors’ tendency to over-sell their work does a disservice both to their field and, over the longer term, to themselves.

Response:

2a. “Turnkey” comment: The phrase turnkey is meant to describe a commercially available microscope that is accessible to the broad biological community. By that definition (see also the Oxford dictionary definition of turnkey), our approach is turnkey as the imaging does not require any hardware or software modifications. Moreover, there is strong precedent to using this adjective when describing microscopy techniques that can be implemented on a commercial system. See the following examples of recent *Nature Communications* papers that use the term “turnkey” or “turn-key” to describe microscopy and analysis methods:

In Stabley et al. “Real-time fluorescence imaging with 20 nm axial resolution” 2015

“We have developed a simple turn-key strategy based on a commercial TIRF microscope to determine the z-position of proteins within a cell with nanometre resolution in real-time.”

In Brown et al. “Single-molecule detection on a portable 3D-printed microscope” 2019

“However, as discussed in a recent review, the financial barriers to buy and maintain a commercial turn-key microscope are still important for most laboratories⁷. Many groups are developing their own high-end instruments⁸, acquisition hardware, and software, but the blueprints of these setups are often too complex for replication by other groups.”

In Krull et al. “Artificial-intelligence-driven scanning probe microscopy” 2020

“DeepSPM brings state-of-the-art SPM closer to a turnkey application, enabling non-expert users to achieve optimal performance.”

In Eliceiri et al. “Biological imaging software tools” 2012

“The obvious advantage of commercial image-acquisition packages is that they provide a turnkey solution to all ‘standard’ image-analysis strategies (acquiring individual images, taking time-lapse series, collecting three-dimensional (3D) stacks at multiple x-y positions and so on).”

In Levet et al. “A tessellation-based colocalization analysis approach for single-molecule localization microscopy” 2019

“With the widespread of SMLM techniques helping to decipher important biological questions, it becomes crucial to provide access to robust and turn-key analysis methods that can be used by non-experts without biasing the data interpretation.”

2b. digital versus analog tension sensors: The folded DNA probes are threshold “digital” sensors and counterintuitively this is a more precise approach to measure molecular forces compared to analog sensors. This is because an ensemble of two-state sensors will faithfully inform on the absolute number of molecules experiencing force exceeding threshold force. In contrast, analog sensors will not provide this information. Unless one is doing single molecule measurements, analog sensors fail at providing quantitative information on the absolute number density of mechanical events. We address this point in the new table comparing SIM-MFM and TFM

2c. orientational degeneracy:

We have expanded our description of this limitation in the discussion

*“... is unlikely that such acquisitions could be implemented using existing SIM microscope hardware. Finally, a key limitation of MFM is that force orientation measurement is degenerate (i.e. each measured orientation could be one of two orientations that mirror each other across the z-axis). Perhaps eventually, SIM will be performed with **E** dipoles that are not parallel to the coverslip, which would enable nondegenerate force vector mapping¹⁴. Again, such advances will likely not be possible without hardware modifications to existing SIM microscopes.”*

We have also added a description of this limitation to the introductory section of the manuscript:

“...spends ~10% of its time unstacked from the duplex terminus (during which time the fluorophore is randomly oriented)³. Note that the 180° periodicity, as shown in equation (4), means that force orientation is degenerate (i.e. any orientation measurement is indistinguishable from the same orientation rotated 180° around the z-axis). This two-fold degeneracy in force orientation measurement is a fundamental limitation of MFM that could potentially be addressed in the future using inclined illumination approaches¹⁴.”

2d. Spatial resolution:

We believe that this qualified language is appropriate because MTFM does in fact yield information that is more directly related to single molecule forces. We have measured the density of hairpins on our chips at ~200 molecules/micron², and ~1-5% of hairpins are engaged and mechanically unfolded at any given time point. These calculations indicate that MTFM force signal is produced by ~1-10 open hairpins and in SIM-MFM this number is smaller. As a result, SIM-MFM produces averaged signal from a discrete number of probes, which is in contrast to TFM which averages the forces from thousands of receptors per “element”. It is possible to detect the signal from single hairpins experiencing pN forces, and this is not the case for TFM which requires the collective activity of thousands of adhesion receptors. Nonetheless, we have removed the language referenced.

Comment 3: The authors should provide evidence that the SIM maps presented are not subject to reconstruction artifacts. For example, the reticular structures observed in Figure 3e are reminiscent of the artifacts previously noted when using Weiner deconvolution (Fan et al. Biophys Rep. 2019) and low signal intensities.

Response: See response to reviewer 2, comment 6.

Comment 4: The authors report that Monte Carlo simulation suggests that estimates for the polar angle theta are poor at low photon counts—presumably this is why they discard low intensity pixels in their analysis. For this reason, it is important that they quantify and report the distribution of photon counts and the cutoff applied for their analyses.

Response: We appreciate this suggestion and have taken the additional step of reporting the number of photons collected on the EMCCD such that we can compare the modeling to the results. The results of this additional analysis is shown below is now added as a new supplemental figure (Fig. S11). Briefly, we generated histograms of pixel photon counts for a single representative cell and also for $n=17$ cells. The data shows that most accepted signal lies in the range of $300 < I_{max} < 1,000$ photons, while the brightest 10% of signal lies in the range of $1,000 < I_{max} < 3,000$ photons. Therefore, the MFM force orientation measurements are reliable based on the photon intensities collected.

Photon counts were estimated from arbitrary units (a.u.) by 1) subtracting the 200 a.u. baseline from raw images, 2) multiplying by the pre-amplification factor (4.9, also called “Conversion Gain #1” in the nd2 image metadata), and 3) dividing by the conversion gain (100). We then used a masking procedure described in **Fig. S13** to select pixels that were included in SIM-MFM analyses. **a)** A histogram of photon counts following this full process is shown for an individual platelet, along with **b)** a cumulative density function of counts. **c,d)** Same as **a** and **b**, but for an aggregated dataset consisting of 17 platelets. The data shows that most accepted signal lies in the range of $300 < I_{max} < 1,000$ photons, while the brightest 10% of signal lies in the range of $1,000 < I_{max} < 3,000$ photons. Therefore, the MFM force orientation measurements are reliable based on the photon intensities collected.

We have also included a reference to this new figure in the main text:

“Our results at the experimentally relevant signal level of $I_{max} = 1,000$ photons (Fig. S7) displayed small systematic errors in ϕ (less than half a degree across all orientations).”

Comment 5: The polar angle for the force vector of 45 degrees measured for 3T3s seems large relative to previous measurements (Liu PNAS 2015). It seems plausible that this may reflect the tendency of the measurement used here to overestimate low theta values (Fig. S5E). Liu et al. is, incidentally, another relevant prior publication that is not cited.

Response: First, we would like to note that Liu et al. was reporting the angle from the horizontal plane, and we are reporting the angle from the vertical axis. Therefore, we are actually reporting values that are larger than that of Liu et al. Secondly, MFM is more precise in reporting force orientations that are more parallel to the horizontal plane, as these force vectors produce greater amplitude in the fit sinusoids. Finally, we emphasize that there is considerable debate in the literature and other work has reported talin tilt angles that are more consistent with our 45 degree value. For example, see the work of Weaver and Paszek in *Nature Methods* and also Waterman and Springer in *Nature Communications*^{21, 22}.

As a final note, the literature precedent has mostly focused on measuring the tilt of talin and other focal adhesion proteins, while our work reports the tilt of the integrin ligand itself. To the best of our knowledge, these two vectors may not necessarily align due to the complex architecture of adhesion assemblies and biophysical factors including the plasma membrane. For example, it is not clear what rotational and tilt angles the alpha-beta integrin heterodimer can adopt within the focal adhesion. To accurately portray this controversy, we have included a citation to Liu et al. in our updated text (see response to reviewer 2, comment 18).

Comment 6: The rationale for splitting platelets into “two groups (28 increasing alignment cells and 22 cells with non-increasing alignment)” is not clear. The authors should provide evidence that these cells are distinguishable in an independent, biologically relevant way. Otherwise, the parsimonious interpretation is that all the cells come from the same distribution, of which half happen to fall above the measurement threshold for force vector alignment. This ambiguity makes this section of the paper difficult to evaluate.

Response: We apologize for the lack of clarity in our original submission. We did not mean to suggest that these two groups represent biologically distinct populations of cells. Rather, we binned the cells into two groups based on their phenotype; only the increasing-alignment cells would fit well to equation 9, so it would distort our results if we included non-increasing alignment cells in the calculation of average τ_{align} . Notably, heterogeneity in cell phenotype and even in platelet phenotypes is common. This is documented in platelet textbooks. In our hands, even when we study a homogeneous population of platelet from a single donor and plate these on chemically uniform surfaces, we find a variety of responses, with a subset of platelets displaying a spiky morphology (filopodial projections) while other platelets spreading uniformly on the substrate. Others have reported heterogeneity in platelet cytoskeleton. For example, prior work by Lickert *et al.* presented in *Scientific Reports* found that human platelets spreading on fibronectin-coated glass showed actin and vinculin was generally concentrated in 2, 3, or 4 lobes (65%, 19%, or 3% of platelets, respectively), while the remainder of platelet (13%) were isotropically (e.g. circularly) organized⁶. Accordingly, classifying the platelets in two groups based on the dynamics of their tension signal is consistent with past literature.

We have modified our discussion of the grouping to better communicate the purpose behind the separation:

“To quantify the average behavior of the increasing-alignment population, we split these cells into two groups (28 increasing alignment cells and 22 cells with non-increasing alignment) and compared the sets of fit-parameters.”

Furthermore, we have expanded upon our discussion of the classification of the cell phenotype in the end of the relevant results section.

“...underpinning this phenomenon. Our analysis of increasing alignment was restricted to 56% of platelets studied, as the remainder did not appear to exhibit increasing alignment. However, a localization microscopy-based analysis of human platelets spreading on fibronectin-coated glass found that platelet actin and vinculin was generally concentrated in 2, 3, or 4 lobes (65%, 19%, or 3% of platelets, respectively), while the remainder of platelet (13%) were isotropically (e.g. circularly) organized. In this work, we expect that only the 2-lobe subset should be recognized as increasing-alignment. This limitation occurs because, if alignment is increasing internally with 3 or 4 lobes the whole-cell R measurement should appear artificially low and remain largely time-invariant. Therefore, the non-increasing alignment group may include 3- or 4-lobed platelets with alignment that is increasing internally within lobes. As such, we expect that our findings regarding the dynamics of alignment may be applicable to greater than 56% of platelets.”

Comment 7: The null result reported for T-cells (no obvious force polarization) is likewise difficult to interpret. The three possibilities listed by the authors are reasonable. However, it also seems possible that the force threshold for the sensor used may be too low or too high to measure the forces that are most relevant for T-cell activation. Another possibility is that the choice of antibody might matter: tangential force transmitted through 17A2 resulted in T-cell activation, whereas other mABs did not (Kim et al. JBC 2009). (So far as I can tell the authors do not report what antibody they used.) All of these ambiguities make this section of the paper likewise difficult to interpret.

Response: Our prior published work and also the work of Zhu and colleagues^{23, 24} (*Nature Immunology* 2018) has shown TCR activation using antibodies generates tension signal that unfolds the 4.7 pN probes. The 4.7 pN hairpin used here is identical in sequence to the probe used previously^{25, 26} (Liu & Salaita *et al.*, PNAS 2016, Ma & Salaita *et al.*, PNAS 2019). Our previous work paper clearly shows that T cells are activated using this probe and are fully capable of opening probes of this $F_{1/2}$. Therefore, we can rule out this possibility as a potential reason for generating null signal.

Regarding the choice of antibody, we apologize for neglecting to report this important detail. We have updated our manuscript to specify the antibody. Regarding the antiCD3 antibody, we use the clone 145-2C11. This clone binds to the epsilon domain of antiCD3 and is a known activating antibody. Therefore, the antibody will activate T cells regardless of the direction of the force. We have adjusted our manuscript to specify this:

We cultured primary mouse CD8+ T-cells on DNA hairpin tension probes that present antibodies to CD3ε (CD3ε is part of the TCR complex which includes CD3δ, γ and ζ chains, as well as the TCR α/β chains). The antibody is the clone 145-2C11, a known activating antibody that should activate T cells regardless of force orientation. As observed previously^{25, 26}, T-cells spread on the tension probe-functionalized surface, and the TCRs engaged and mechanically unfold tension probes to generate bright tension fluorescence signal.

Comment 8: The method used to derive confidence bands in Figure 5h is ad hoc, and needs to be replaced with a more rigorous approach.

Response: We have replaced the curves' confidences intervals with a bootstrapping-based method.

References

1. Au - Ma, R.; Au - Kellner, A. V.; Au - Hu, Y.; Au - Deal, B. R.; Au - Blanchard, A. T.; Au - Salaita, K. *JoVE*, e62348.
2. Iqbal, A.; Wang, L.; Thompson, K. C.; Lilley, D. M. J.; Norman, D. G. *Biochemistry* **2008**, *47*, (30), 7857-7862.
3. Iqbal, A.; Arslan, S.; Okumus, B.; Wilson, T. J.; Giraud, G.; Norman, D. G.; Ha, T.; Lilley, D. M. J. *Proc. Natl. Acad. Sci. U. S. A.* **2008**, *105*, (32), 11176.
4. Brockman, J. M.; Blanchard, A. T.; Pui-Yan Ma, V.; Derricotte, W. D.; Zhang, Y.; Fay, M. E.; Lam, W. A.; Evangelista, F. A.; Mattheyses, A. L.; Salaita, K. *Nat. Methods* **2017**, *15*, 115.
5. Basu, R.; Whitlock, B. M.; Husson, J.; Le Floc'h, A.; Jin, W.; Olyer-Yaniv, A.; Dotiwala, F.; Giannone, G.; Hivroz, C.; Biais, N.; Lieberman, J.; Kam, L. C.; Huse, M. *Cell* **2016**, *165*, (1), 100-110.
6. Lickert, S.; Sorrentino, S.; Studt, J.-D.; Medalia, O.; Vogel, V.; Schoen, I. *Sci. Rep.* **2018**, *8*, (1), 5428.
7. Haddad, E.; Cramer, E.; Rivière, C.; Rameau, P.; Louache, F.; Guichard, J.; Nelson, D. L.; Fischer, A.; Vainchenker, W.; Debili, N. *Blood* **1999**, *94*, (2), 509-518.
8. Sakurai, Y.; Fitch-Tewfik, J. L.; Qiu, Y.; Ahn, B.; Myers, D. R.; Tran, R.; Fay, M. E.; Ding, L.; Spearman, P. W.; Michelson, A. D.; Flaumenhaft, R.; Lam, W. A. *Blood* **2015**, *126*, (4), 531-8.
9. Hanke, J.; Ranke, C.; Perego, E.; Köster, S. *Soft Matter* **2019**, *15*, (9), 2009-2019.
10. Zhang, Y.; Qiu, Y.; Blanchard, A. T.; Chang, Y.; Brockman, J. M.; Ma, V. P.; Lam, W. A.; Salaita, K. *Proc. Natl. Acad. Sci. U. S. A.* **2017**.
11. Niekamp, S.; Stuurman, N.; Vale, R. D. *Nat. Methods* **2020**, *17*, (4), 437-441.
12. Aitken, C. E.; Marshall, R. A.; Puglisi, J. D. *Biophys. J.* **2008**, *94*, (5), 1826-1835.
13. Zhanghao, K.; Chen, X.; Liu, W.; Li, M.; Liu, Y.; Wang, Y.; Luo, S.; Wang, X.; Shan, C.; Xie, H.; Gao, J.; Chen, X.; Jin, D.; Li, X.; Zhang, Y.; Dai, Q.; Xi, P. *Nat. Commun.* **2019**, *10*, (1), 4694.
14. Blanchard, A. T.; Brockman, J. M.; Salaita, K.; Mattheyses, A. L. *Opt. Express* **2020**, *28*, (7), 10039-10061.
15. Glazier, R.; Brockman, J. M.; Bartle, E.; Mattheyses, A. L.; Destaing, O.; Salaita, K. *Nat. Commun.* **2019**, *10*, (1), 4507.
16. Kanchanawong, P.; Shtengel, G.; Pasapera, A. M.; Ramko, E. B.; Davidson, M. W.; Hess, H. F.; Waterman, C. M. *Nature* **2010**, *468*, (7323), 580-262.
17. Brockman, J. M.; Su, H.; Blanchard, A. T.; Duan, Y.; Meyer, T.; Quach, M. E.; Glazier, R.; Bazrafshan, A.; Bender, R. L.; Kellner, A. V.; Ogasawara, H.; Ma, R.; Schueder, F.; Peitrich, B. G.; Jungmann, R.; Li, R.; Mattheyses, A. L.; Ke, Y.; Salaita, K. *Nat. Methods* **2020**, *17*, (10).
18. Hu, S.; Tee, Y.-H.; Kabla, A.; Zaidel-Bar, R.; Bershadsky, A.; Hersen, P. *Cytoskeleton* **2015**, *72*, (5), 235-245.
19. Stubb, A.; Laine, R. F.; Miihkinen, M.; Hamidi, H.; Guzmán, C.; Henriques, R.; Jacquemet, G.; Ivaska, J. *Nano Lett.* **2020**, *20*, (4), 2230-2245.
20. Brockman, J. M.; Su, H.; Blanchard, A. T.; Duan, Y.; Meyer, T.; Quach, M. E.; Glazier, R.; Bazrafshan, A.; Bender, R. L.; Kellner, A. V.; Ogasawara, H.; Ma, R.; Schueder, F.; Petrich, B. G.; Jungmann, R.; Li, R.; Mattheyses, A. L.; Ke, Y.; Salaita, K. *Nat. Methods* **2020**.
21. Paszek, M. J.; DuFort, C. C.; Rubashkin, M. G.; Davidson, M. W.; Thorn, K. S.; Liphardt, J. T.; Weaver, V. M. *Nat. Methods* **2012**, *9*, 825.
22. Nordenfelt, P.; Moore, T. I.; Mehta, S. B.; Kalappurakkal, J. M.; Swaminathan, V.; Koga, N.; Lambert, T. J.; Baker, D.; Waters, J. C.; Oldenbourg, R.; Tani, T.; Mayor, S.; Waterman, C. M.; Springer, T. A. *Nat. Commun.* **2017**, *8*, (1), 2047.
23. Zhu, C.; Chen, W.; Lou, J.; Rittase, W.; Li, K. *Nat. Immunol.* **2019**, *20*, (10), 1269-1278.

24. Hong, J.; Ge, C.; Jothikumar, P.; Yuan, Z.; Liu, B.; Bai, K.; Li, K.; Rittase, W.; Shinzawa, M.; Zhang, Y.; Palin, A.; Love, P.; Yu, X.; Salaita, K.; Evavold, B. D.; Singer, A.; Zhu, C. *Nat. Immunol.* **2018**, *19*, (12), 1379-1390.
25. Liu, Y.; Blanchfield, L.; Ma, V. P.-Y.; Andargachew, R.; Galior, K.; Liu, Z.; Evavold, B.; Salaita, K. *Proc. Natl. Acad. Sci. U. S. A.* **2016**, *113*, (20), 5610-5615.
26. Ma, R.; Kellner, A. V.; Ma, V. P.-Y.; Su, H.; Deal, B. R.; Brockman, J. M.; Salaita, K. *Proc. Natl. Acad. Sci. U. S. A.* **2019**, *116*, (34), 16949.

Reviewers' Comments:

Reviewer #1:

Remarks to the Author:

I thank the authors for thoroughly revising the manuscript. During the revision process, the authors have addressed all pertinent questions. I have no more queries on the revised version.

I am looking forward to the implementation of DNA tension probes in SIM-MFM based on the approach developed by Blanchard et al., by a broader user base across cellular mechanobiology and beyond.

Yours sincerely,

Raghavendra Palnakar, PhD

Reviewer #2:

Remarks to the Author:

The authors have addressed my previous concerns adequately. Moreover, they have uploaded their protocols in JoVE, which I believe will add the influence of this technique significantly. Also, in term of the super-resolution part, the authors have detuned their claims to avoid misunderstanding. Hence, I have no further questions, and I am supportive to the publication of the manuscript in its current form.

Reviewer #3:

Remarks to the Author:

The manuscript is substantially improved. The remaining concern is that the practical as opposed to theoretical spatial and angular resolutions of the technique are still difficult to pull out of the text. For example, in table S2, the listed resolution is 110 nm. However, the data provided in Figure S6H suggest a practical resolution of ~210 nm, perhaps due to the limited signal-to-noise that the authors now describe. The authors need to better demonstrate how finite signal-to-noise affects the quality of their spatial and angular maps.

Reviewer 3:

Reviewer comment: The manuscript is substantially improved. The remaining concern is that the practical as opposed to theoretical spatial and angular resolutions of the technique are still difficult to pull out of the text. For example, in table S2, the listed resolution is 110 nm. However, the data provided in Figure S6H suggest a practical resolution of ~210 nm, perhaps due to the limited signal-to-noise that the authors now describe. The authors need to better demonstrate how finite signal-to-noise affects the quality of their spatial and angular maps.

Response: We appreciate the Reviewer's concern and, as such, have included an additional analysis that supports our original resolution estimate of ~110 nm. This new analysis utilizes a recently-published algorithm for resolution estimation¹. We ran 12 pairs of SIM-MFM images (widefield and super-resolution) of platelets through this (parameter-free) algorithm and observed a spatial resolution of 193 ± 8 nm in widefield and 114 ± 2 nm in super-resolution. This analysis is summarized in a new supplemental figure (Fig. S7), shown on the following page. We have also added a brief description of this analysis and a reference to Fig. S7 to the main text:

"... thus validating the quality of the data and reconstruction process. As a second means of directly quantifying resolution, we used parameter-free image resolution analysis software¹. We ran 12 pairs of SIM-MFM images (widefield and super-resolution) through this software and observed a spatial resolution of 193 ± 8 nm in widefield and 114 ± 2 nm in super-resolution (Fig. S7).

We next used SIM-MFM..."

These findings are consistent with our existing findings obtained using linescan-based analysis (Fig. S6). The 210 nm metric referenced by the Reviewer was the thickness of the tension zone at the platelet lamellipodial edge, as measured using SIM reconstruction. Due to non-zero thickness of these edges (recently measured as ~100 nm via high-resolution DNA-PAINT), the observed 210 nm thickness likely arises from adding the tension signal's true thickness (~100 nm) to the resolution of super-resolution SIM-MFM (~110 nm). In light of this confirmation of our previous analysis, we believe the existing discussion and computational simulations of the relationship between noise and angular accuracy (Figs. 4, S10-11) are sufficient.

Figure S7: Image resolution estimation using parameter-free decorrelation analysis

a) Representative SIM-MFM I_{max} images of human platelets, shown in widefield (left – a 512x512 image with 41.7 nm pixel size) and super-resolution (right – a 1024x1024 SIM reconstruction with 20.85 nm pixel size). **b)** Corresponding output plots of images shown in **a** processed using decorrelation analysis software¹. According to ref. 4: green line denotes decorrelation functions before high-pass filtering; magenta line denotes the radial average of log of absolute value of Fourier transform of image; gray lines denote all high-pass filtered decorrelation functions; blue to black lines denote decorrelation functions with refined mask radius and high-pass filtering range; blue crosses denote all local maxima; vertical lines denote cut-off frequency. Resolution is 2 multiplied by the pixel size divided by the cutoff frequency. For more information, see ref. 4. **c)** Boxplots showing the estimated resolution of twelve pairs of images in widefield and super-resolution. Through this analysis, we observed a spatial resolution of $193 \pm 8 \text{ nm}$ (mean \pm standard deviation) in widefield and $114 \pm 2 \text{ nm}$ in super-resolution.

References

1. Descloux, A.; Großmayer, K. S.; Radenovic, A. *Nat. Methods* **2019**, 16, (9), 918-924.

Reviewers' Comments:

Reviewer #3:

Remarks to the Author:

The additional analysis of the spatial resolution of the authors' measurements is a welcome addition to the manuscript. I now view it as ready for publication.

REVIEWERS' COMMENTS

Reviewer #3 (Remarks to the Author):

The additional analysis of the spatial resolution of the authors' measurements is a welcome addition to the manuscript. I now view it as ready for publication.

Response:

We thank the reviewer for their time and commitment to improving this manuscript.